# GLP-1R signaling neighborhoods associate with the susceptibility to adverse drug reactions of incretin mimetics

Shane C. Wright [1,2,3,14,15] ✉, Aikaterini Motso[1,14], Stefania Koutsilieri[1], Christian M. Beusch [4], Pierre Sabatier [4,5,6], Alessandro Berghella [7,8], Élodie Blondel-Tepaz[2,3], Kimberley Mangenot[2,3], Ioannis Pittarokoilis [1], Despoina-Christina Sismanoglou[1], Christian Le Gouill [2], Jesper V. Olsen [5], Roman A. Zubarev [4,9,10], Nevin A. Lambert [11], Alexander S. Hauser [7], Michel Bouvier [2,15] ✉ & Volker M. Lauschke [1,12,13,15] ✉

G protein-coupled receptors are important drug targets that engage and activate signaling transducers in multiple cellular compartments. Delineating therapeutic signaling from signaling associated with adverse events is an important step towards rational drug design. The glucagon-like peptide-1 receptor (GLP-1R) is a validated target for the treatment of diabetes and obesity, but drugs that target this receptor are a frequent cause of adverse events. Using recently developed biosensors, we explored the ability of GLP-1R to activate 15 pathways in 4 cellular compartments and demonstrate that modifications aimed at improving the therapeutic potential of GLP-1R agonists greatly influence compound efficacy, potency, and safety in a pathway- and compartment-selective manner. These findings, together with comparative structure analysis, time-lapse microscopy, and phosphoproteomics, reveal unique signaling signatures for GLP-1R agonists at the level of receptor conformation, functional selectivity, and location bias, thus associating signaling neighborhoods with functionally distinct cellular outcomes and clinical consequences.

As clinicians, healthcare systems, and the pharmaceutical industry try to move away from generic disease management towards precision medicine, several fundamental issues need to be addressed in order to maximize therapeutic efficacy and minimize adverse events[1]. Among these are the mutational and epigenetic landscape of drug targets, polypharmacology and an incomplete knowledge of functional selectivity or biased signaling. In particular, the functional selectivity of existing compounds that target G protein-coupled receptors (GPCRs)

[1]Department of Physiology & Pharmacology, Karolinska Institutet, Stockholm, Sweden. [2]Institute for Research in Immunology and Cancer, Université de Montréal, Montréal, QC H3T 1J4, Canada. [3]Department of Biochemistry and Molecular Medicine, Université de Montréal, Montréal, QC H3T 1J4, Canada. [4]Chemistry I, Department of Medical Biochemistry and Biophysics, Karolinska Institute, Stockholm 17177, Sweden. [5]Novo Nordisk Foundation Centre for Protein Research, Faculty of Health and Medical Sciences, University of Copenhagen, 2200 Copenhagen, Denmark. [6]Department of Surgical Sciences, Uppsala University, Uppsala 75185, Sweden. [7]Department of Drug Design and Pharmacology, Faculty of Health and Medical Sciences, University of Copenhagen, Copenhagen, Denmark. [8]Faculty of Bioscience and Agro-Food and Environmental Technology, University of Teramo, Teramo 64100, Italy. [9]Department of Pharmacological & Technological Chemistry, I.M. Sechenov First Moscow State Medical University, Moscow 119146, Russia. [10]The National Medical Research Center for Endocrinology, Moscow 115478, Russia. [11]Department of Pharmacology and Toxicology, Medical College of Georgia, Augusta University, Augusta, GA 30912, USA. [12]Dr Margarete Fischer-Bosch Institute of Clinical Pharmacology, Stuttgart, Germany. [13]University of Tübingen, Tübingen, Germany. [14]These authors contributed equally: Shane C. Wright, Aikaterini Motso. [15]These authors jointly supervised this work: Shane C. Wright, Michel Bouvier, Volker M. Lauschke. ✉e-mail: shane.wright@ki.se; michel.bouvier@umontreal.ca; volker.lauschke@ki.se

−the largest family of drug targets−remains largely unexplored and is often limited to a couple of downstream, signal-amplified readouts[2]. The dynamic nature of GPCR architecture enables diversity across ligand binding pockets and intracellular binding sites−increasing the number of pathways that can be engaged[3–5]. This inherent diversity has implications for rational drug design, which aims to balance efficacy and adverse drug reactions (ADRs).

In recent years, the glucagon-like peptide-1 receptor (GLP-1R) has emerged as an important target in the treatment of type II diabetes and obesity[6,7]. Biologics that target this receptor, such as semaglutide and liraglutide have proven clinically successful in regulating blood glucose levels, but they are not without drawbacks that include daily or weekly injections and reports of ADRs ranging from nausea and diarrhea to pancreatitis[7]. In order to produce an orally available GLP-1R agonist, increased efforts have been directed towards the development of small molecules. Incretin mimetics engage both the extracellular and transmembrane domains (TMD) of GLP-1R, whereas small molecule GLP-1R agonists are more restricted to the traditional GPCR binding pocket in the TMD[8–11]. Differences in binding pose have been shown to lead to unique signatures in transducer engagement for which emphasis has been placed on $G_s$ and β-arrestin (βARR). Recent work from our group demonstrated that the endogenous agonist for GLP-1R, GLP-1 (7-36), activates members from all G protein families at the plasma membrane[5]. However, our knowledge about the signaling signatures of preclinical and clinical GLP-1R agonists and the location of this activity is incomplete.

The ability of GPCRs to signal from different subcellular compartments is now well-established[12–21] and there is convincing evidence that subcellular signaling can lead to ADRs[22–25]. In light of the increasing attention that GLP-1R has received as a drug target, we sought to better characterize the signaling fingerprint of this receptor after treatment with preclinical and clinical GLP-1R agonists. To this end, we developed an integrative and comprehensive approach to monitor transducer promiscuity with subcellular resolution based on bioluminescence resonance energy transfer (BRET). This strategy allowed us to decode the specific transducer recruitment signatures of GLP-1R at the plasma membrane, early endosomes, the Golgi apparatus, and the endoplasmic reticulum and the resulting signaling fingerprints correlated with ADR risk in the FDA Adverse Event Reporting System (FAERS). At the molecular level, differences in transducer engagement entailed altered kinetics and connectivity of phosphoproteomic signaling networks. These findings open up new possibilities for fine-tuning drug development and suggest that preclinical pipelines to assess efficacy and safety should be reassessed to ensure that the benefit to the patient outweighs any risk.

## Results

### GLP-1R agonists differentially engage residue contact networks

The rapid development of incretin mimetics has resulted in several inactive and active-state ligand-GLP-1R structures that have been solved by cryo-EM or X-ray crystallography. Comparative analysis of these structures can offer insight into the conformational plasticity of GLP-1R as a basis for investigating functional selectivity. To better understand the drug-specific differences that populate the most common conformer, we analyzed the residue contact networks of several biologic- and small molecule-bound structures for GLP-1R that encompassed both inactive and active conformations. In this analysis, we included the negative allosteric modulators NNC0640 (PDB: 5VEX) and PF-06372222 (PDB: 6KJV), the peptide agonists GLP-1 (7-36) (PDB: 6X18), exenatide (PDB: 7LLL) and semaglutide (PDB: 7KI0) as well as the small molecule agonist danuglipron (PDB: 6X1A) (Supplementary Data 1). Contact pairs were grouped across ligand-bound structures to uncover pairs that were either shared across all structures (universal) or limited to subgroups, such as active, inactive, biologic-bound or drug-specific structures (Fig. 1a). Receptor

mapping of these contact pairs revealed that transducer binding resulted in a network of contacts at the lower half of GLP-1R shared across agonist-bound structures. Universal, inactive-state, biologic-bound and ligand-specific contact pairs were distributed throughout the receptor structure. Of these contact pairs, intrahelical contacts were more common in ligand-specific structures compared to other groups which harbored more interhelical contacts (Fig. 1b). Interestingly, the proportion of contact pairs resulting from hydrophobic interactions relative to ionic and aromatic interactions was higher in the case of semaglutide-bound GLP-1R compared to other ligand-bound structures. GLP-1 (7-36) and exenatide had a greater proportion of ionic and aromatic interactions, whereas danuglipron had fewer ionic interactions with increased polar interactions (Fig. 1c). Given the unique differences in the number and type of contact pairs across agonist-bound GLP-1R structures, we hypothesized that conformational flexibility would translate into differences in transducer coupling signatures.

### GLP-1R agonists have distinct signaling profiles at the plasma membrane

Based on the structural differences observed in the residue contact analysis, we selected a series of homologous peptide agonists (GLP-1 and exendin-4 analogs) in addition to the small molecule agonist (Supplementary Data 2) to investigate the pathway selectivity of GLP-1R at the plasma membrane using the effector membrane translocation assay (EMTA) based on enhanced bystander BRET (ebBRET) in living cells[5]. Specifically, we included different cleavage products of preproglucagon [glucagon, GLP-1 (1-37) and GLP-1 (7-36)], the GLP-1 analogues liraglutide and semaglutide, synthetic GLP-1 mimetics from the reptile *Heloderma suspectum* (exenatide and lixisenatide) and the small molecule GLP-1R agonist danuglipron.

Due to the cytosolic localization of luciferase-tagged ebBRET biosensors under basal conditions, pathway activation at the plasma membrane can be monitored through co-expression of an energy acceptor (rGFP) anchored to the plasma membrane using the CAAX box from KRAS. Using this approach, translocation of the biosensor from the cytosol to the plasma membrane results in energy transfer, which is indicative of pathway activation[5]. We exploited this methodology to systematically examine transducer engagement for 15 different pathways ($G_s$, $G_q$, $G_{11}$, $G_{14}$, $G_{15}$, $G_{12}$, $G_{13}$, $G_{i1}$, $G_{i2}$, $G_{i3}$, $G_{oA}$, $G_{oB}$, $G_z$, G protein-coupled receptor kinase (GRK)/Gβγ and βARR (Fig. 2a and Supplementary Fig. 1). As expected, GLP-1 (1-37) was not active in any pathway measured; however, its cleavage product, GLP-1 (7-36), activated all tested G proteins (apart from $G_{12}$ and $G_{14}$), GRK/Gβγ and βARR (Fig. 2b). Surprisingly, glucagon, which is used to treat severe hypoglycemia by targeting the glucagon receptor and is a weak agonist for GLP-1R[26], was also found to activate a subset of the pathways elicited by GLP-1 (7-36) stimulation[5], underlining the importance of investigating polypharmacology. GLP-1 (7-36), liraglutide, and exenatide strongly activated members of the $G_{i/o}$ family compared to other tested drugs. Semaglutide and lixisenatide strongly engaged the $G_s$, GRK/Gβγ, and βARR pathways at the plasma membrane, while danuglipron, a small molecule agonist in clinical trials, displayed a unique fingerprint of pathway engagement that appeared to be biased away from βARR towards $G_s$ and $G_{i/o}$. We then quantified the similarity and diversity of transducer engagement across drugs using the Jaccard index for efficacy and potency (Fig. 2c−f and Supplementary Figs. 2, 3). These results demonstrate that different drugs that engage the same target have qualitatively and quantitatively distinct signaling fingerprints at the plasma membrane.

### Peptide-activated GLP-1R traffics to vesicles and the Golgi apparatus

Once activated, agonist-bound GLP-1R is known to leave the plasma membrane through endocytosis resulting in receptor recycling or

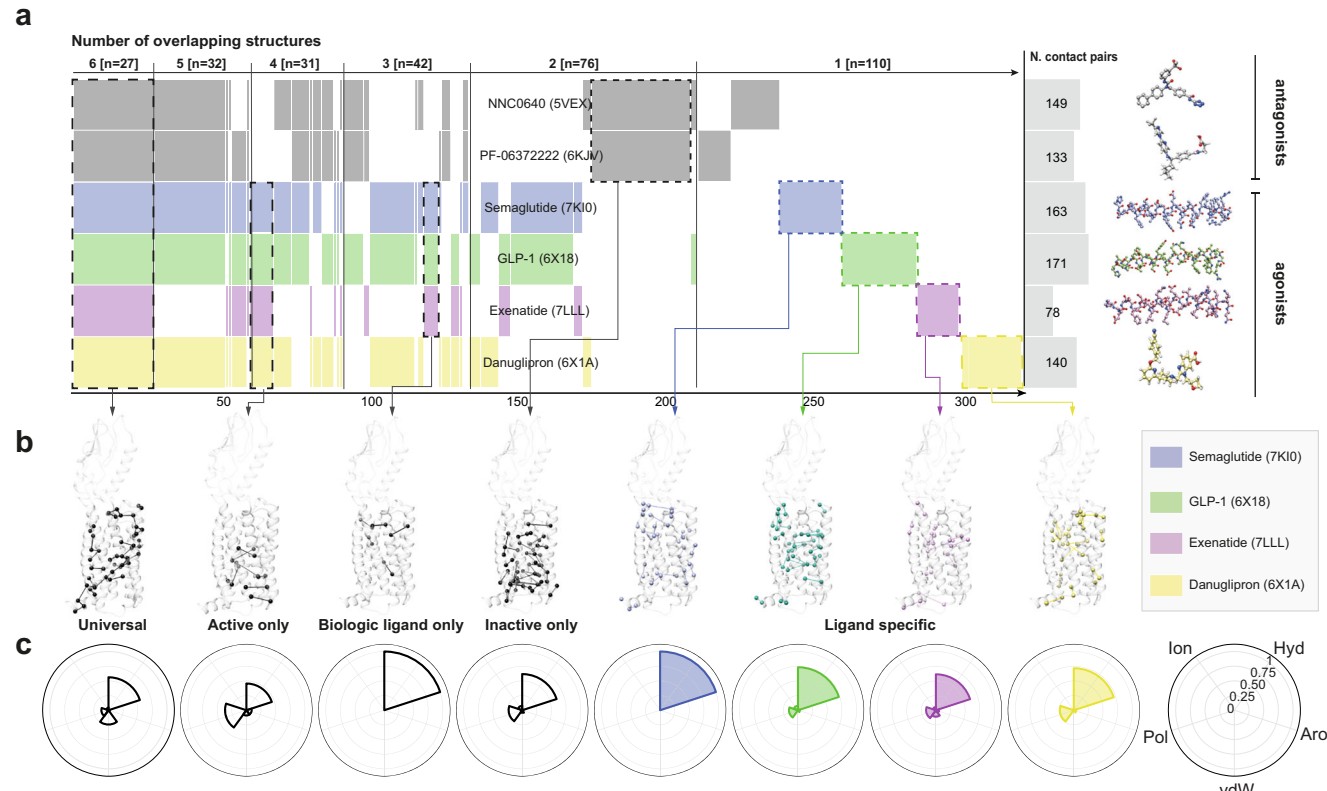

**Fig. 1 | Comparative structure analysis identifies differences in residue contact pairs for GLP-1R agonists. a** Shared contact pairs based on comparative structure analysis[3,62] represented by a Venn-like diagram (*Supervenn*), where each row is a set of PDB-specific contact pairs, and the overlapping parts (groupings) correspond to shared residue contact pairs among a set, sorted by the number of overlapping structures [from six structures (left) to a single structure (right)]. The columns on the right represent the total set sizes (number of residue contact pairs), and the colored ligand structures. The *x*-axis numbers represent a reference scale as a proportion of the total number of residue contacts shared among all structure combinations (*n* = 318). **b** Group-specific and unique shared residue contacts mapped onto a representative GLP-1R structure [PDB: 5NX2[75]]. Residues are denoted as circles (Cα) and the noncovalent contacts between residues are denoted as lines. **c** Polar area diagrams for types of residue contacts (hydrophobic, aromatic, van der Waals, polar and ionic) across ligand-bound structures.

degradation[27]. Receptor trafficking or subcellular localization of GPCRs has functional implications for drug action. In particular, endosomal GLP-1R has been proposed to contribute to the generation of cAMP following receptor activation[28]. However, sustained signaling from internalized GLP-1R does not promote further insulin release and has instead been linked to gastrointestinal ADRs[29,30]. We sought to track active GLP-1R by confocal microscopy using the fluorescent biosensor mini Gs (mGs) that detects the receptor conformation that binds Gs[31]. Using this tool, we could monitor the trafficking dynamics and intracellular signaling neighborhoods that accommodate signaling (Fig. 3a). To this end, we made use of the biologic semaglutide, which displayed the highest efficacy and potency for Gs signaling at the plasma membrane. Exposure of living cells expressing GLP-1R to semaglutide resulted in rapid recruitment of mGs to the plasma membrane (Fig. 3b). This was followed by the time-dependent appearance of mGs-positive vesicles that were positive for the endosomal marker FYVE (Fig. 3c). To our surprise, we also observed a convergence and accumulation of mGs-positive vesicles to the Golgi apparatus suggesting that GLP-1R may engage in endosome-to-Golgi retrograde transport while continuing to sample a conformation that can elicit additional waves of signaling (Fig. 3d, Supplementary Fig. 4 and Supplementary Movie 1). The movement of mGs coincided with the localization of GLP-1R as evidenced by BRET and confocal microscopy demonstrating time-dependent receptor trafficking from the plasma membrane to early endosomes and the Golgi apparatus following agonist stimulation (Supplementary Figs. 5, 6).

## Stimulation of GLP-1R with peptide and small molecule agonists results in selective subcellular activation

Building on these observations, we wondered if endosome-to-Golgi retrograde transport was a common mechanism for peptide and small molecule GLP-1R agonists. To this end, we used the ebBRET platform due to its increased scalability and improved signal-to-noise compared to confocal microscopy to explore drug-induced, compartmentalized pathway engagement for Gs (Fig. 4a). Subcellular resolution was achieved by using organelle-specific energy acceptors for early endosomes, the Golgi apparatus and the endoplasmic reticulum. Interestingly, danuglipron did not promote mGs recruitment at early endosomes at the concentrations tested, but induced a strong response at the Golgi apparatus and was the only drug capable of recruiting mGs to the endoplasmic reticulum (Fig. 4b and Supplementary Fig. 7). From these data, we infer that the small molecule danuglipron can access subcellular pools of GLP-1R in a mechanism that is distinct from peptide-based drugs.

## Changes in the phosphoproteome depend on functional selectivity and location bias

In light of the location bias that was observed for the small molecule relative to peptide agonists of GLP-1R, we sought to examine the functional consequences of differences in intracellular signal transduction by phosphoproteomics. To this end, GLP-1R-expressing cells were treated with danuglipron or semaglutide and phosphoproteomic signatures were evaluated in time series experiments (Fig. 5a and Supplementary Data 3–7). Drug-specific phosphorylation signatures

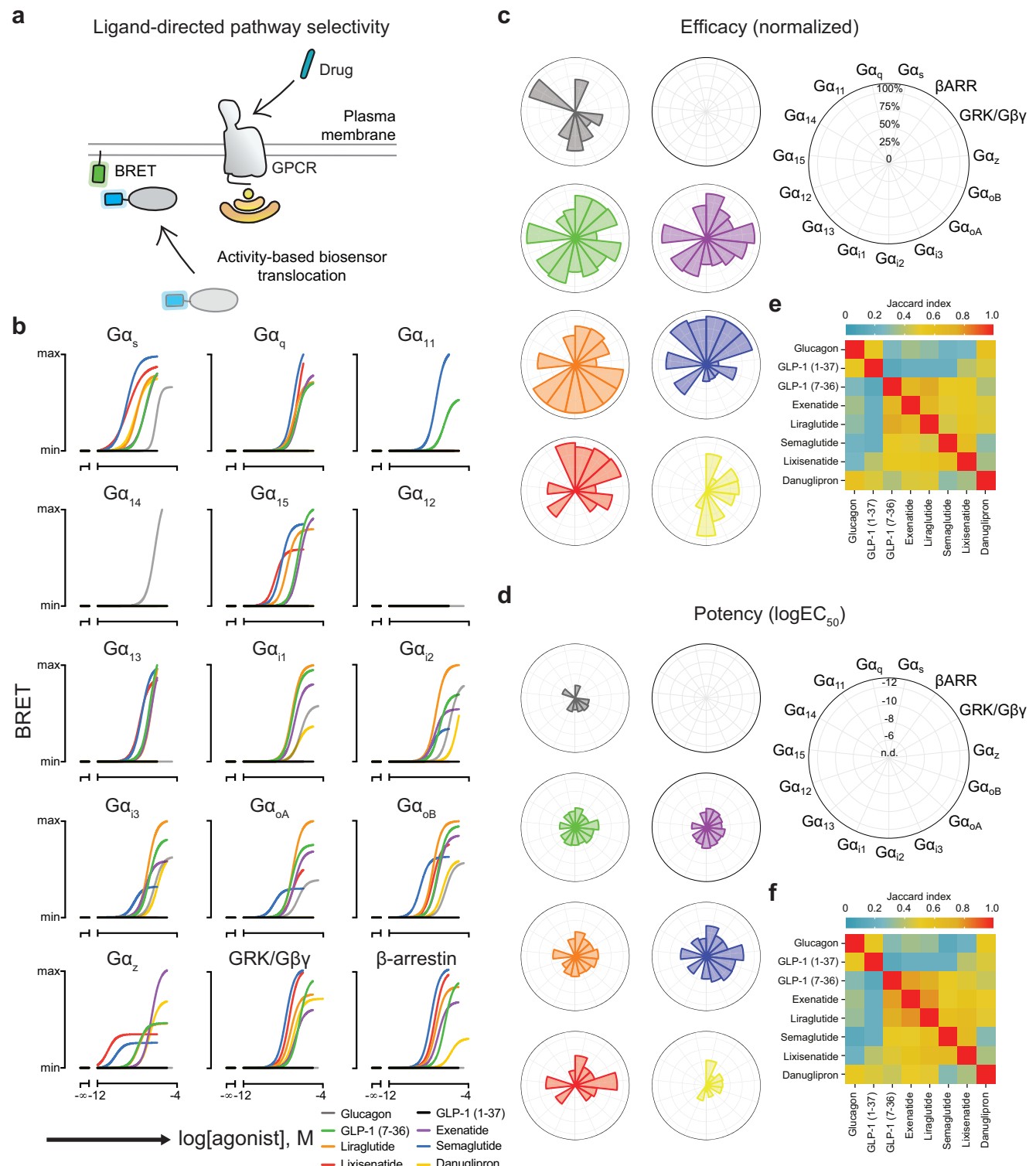

**Fig. 2 | Pharmacological characterization of GLP-1R agonist signaling profiles at the plasma membrane. a** Illustration depicting the ebBRET approach to monitoring compartmentalized signaling. In brief, donor-tagged, pathway-selective biosensors are co-expressed with acceptor-tagged, compartment-specific markers and the GPCR of interest is stimulated with an agonist to monitor pathway activation by measuring BRET. **b** Concentration-response curves of GLP-1R agonists across 15 pathways monitoring biosensor recruitment to the plasma membrane using rGFP-CAAX. Data are represented by the nonlinear fit and scaled according to the highest responding drug ($n = 3$-6 biologically independent experiments). Drug-specific polar area diagrams for efficacy (normalized to the highest responding drug) (**c**) and potency (logEC$_{50}$) (**d**) of 15 pathways at the plasma membrane. Drugs were deemed to activate a given pathway after comparing the top and bottom parameters from nonlinear regression by one-sided extra sum-of-squares F test followed by Bonferroni correction for 8 compounds ($P < 0.00625$). Jaccard similarity index for efficacy (**e**) and potency (**f**) quantifies similarities and differences across drug responses.

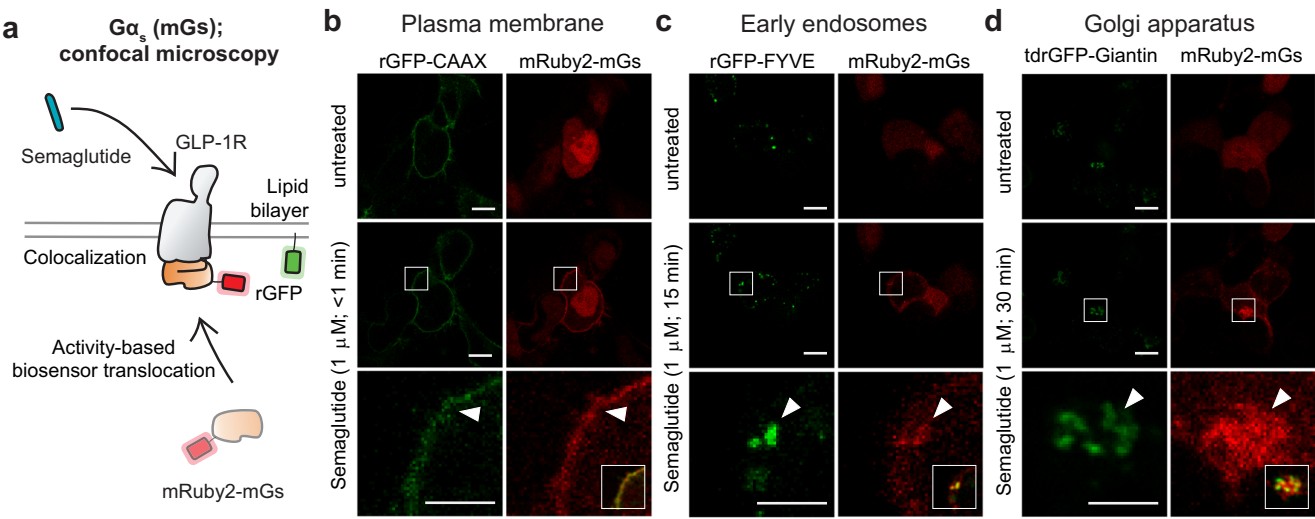

**Fig. 3 | Peptide-activated GLP-1R traffics to vesicles and the Golgi apparatus.**
**a** Schematic of the live cell imaging experiment monitoring agonist-induced mRuby2-mGs translocation to different cellular compartments where rGFP is expressed (PM – rGFP-CAAX; EE – rGFP-FYVE; GA – tdrGFP-Giantin). **b** Confocal images of HEK 293 cells expressing GLP-1R, mRuby2-mGs, and rGFP-CAAX and exposed to semaglutide (1 μM) for less than 1 min. mRuby2-mGs is recruited to active GLP-1R at the plasma membrane (closed arrowhead). **c** Confocal images of HEK293 cells expressing GLP-1R, mRuby2-mGs and rGFP-FYVE and exposed to

semaglutide (1 μM) for 15 min. mRuby2-mGs is recruited to active GLP-1R at early endosomes (closed arrowhead). **d** Confocal images of HEK293 cells expressing GLP-1R, mRuby2-mGs and tdrGFP-Giantin and exposed to semaglutide (1 μM) for 30 min. mRuby2-mGs is recruited to active GLP-1R at the Golgi apparatus (closed arrowhead). Images are representative of three independent experiments. Insets depict merged images of the semaglutide-treated condition. (Scale bars, 10 μm [top and middle] and 5 μm [bottom].) PM plasma membrane, EE early endosomes, GA Golgi apparatus.

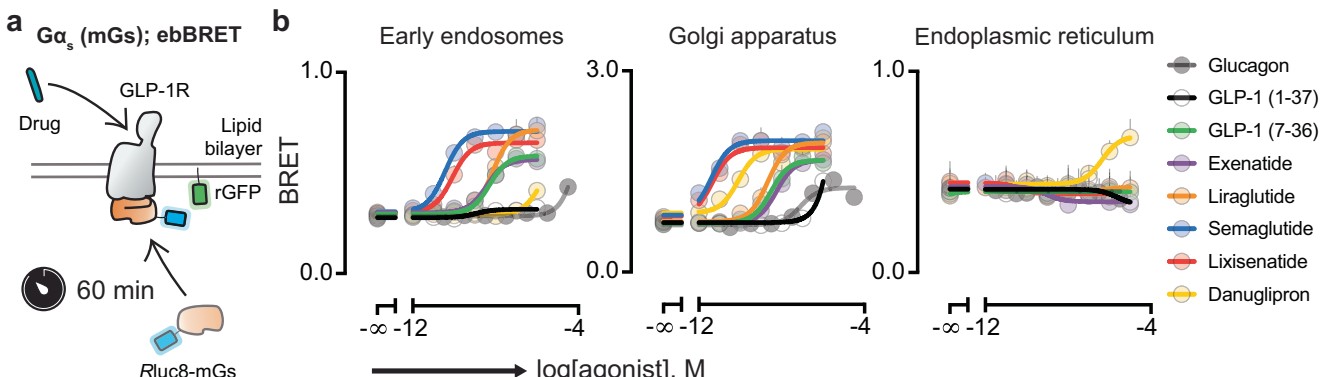

**Fig. 4 | Stimulation of GLP-1R with peptide and small molecule agonists results in selective subcellular activation. a** Schematic of the live cell experiment monitoring agonist-induced Rluc8-mGs translocation to different cellular compartments where rGFP is expressed (EE – rGFP-FYVE; GA – tdrGFP-Giantin; ER – tdrGFP-

PTP1B) based on BRET. **b** Concentration-response curves for $G_s$ pathway activation of GLP-1R agonists at early endosomes, Golgi apparatus and endoplasmic reticulum. Data are represented as the mean ± SEM ($n = 3$ biologically independent experiments). EE early endosomes, GA Golgi apparatus, ER endoplasmic reticulum.

began to appear between 5 and 30 min after exposure, the magnitude of these changes differing between treatments (Fig. 5b). This is consistent with our observations related to signal propagation and compartmentalized GLP-1R activity by BRET and confocal microscopy. Danuglipron-treated cells at 30 min harbored the greatest number of significantly altered phosphopeptides (220 compared to 48 in the semaglutide-specific group with only 8 peptides overlapping between the two treatments) (Fig. 5c). These changes were receptor-mediated and sensitive to an inhibitor of internalization (Fig. 5d). Intriguingly, little overlap in phosphopeptides with significantly altered abundance was observed across timepoints underlining the dynamic and transient nature of these signaling cascades. In addition to detecting phosphoregulation of proteins downstream of $G_s$ like filamin A[32] and GSK3A[33], we also detected evidence of pathway-specific activity like AP2 by βARR[34], AKAP5 by $G_s$/$G_q$[35], ARHGEF11 by $G_i$/$G_{13}$[36, 37], AKAP13 by $G_s$/$G_q$/$G_{13}$[38] and trafficking related proteins like GBF1, RAB1B and α-

synuclein (Supplementary Fig. 8). In particular, RAB1B and α-synuclein[39, 40], proteins involved in ER to Golgi transport, were significantly regulated by danuglipron, but not semaglutide. This is interesting as danuglipron was the only compound capable of engaging mGs in the endoplasmic reticulum. Combined, these results suggest that danuglipron stimulation may feed into the anterograde transport of nascent GLP-1R. In contrast, semaglutide-induced phosphorylation of GBF1, a *cis*-Golgi-localized GEF[41], fits nicely with the observation that peptide-bound GLP-1R can participate in endosome-to-TGN transport. Next, we performed kinase-substrate enrichment analyses to identify signaling nodes that could be connected to peptide- and small molecule-specific GLP-1R transducers and their signaling neighborhoods. While kinase activity was generally conserved across drug treatments, the magnitude (fold enrichment compared to vehicle) and trend of substrate enrichment varied greatly (Fig. 5e). Importantly, kinase kinetics differed between small molecule and

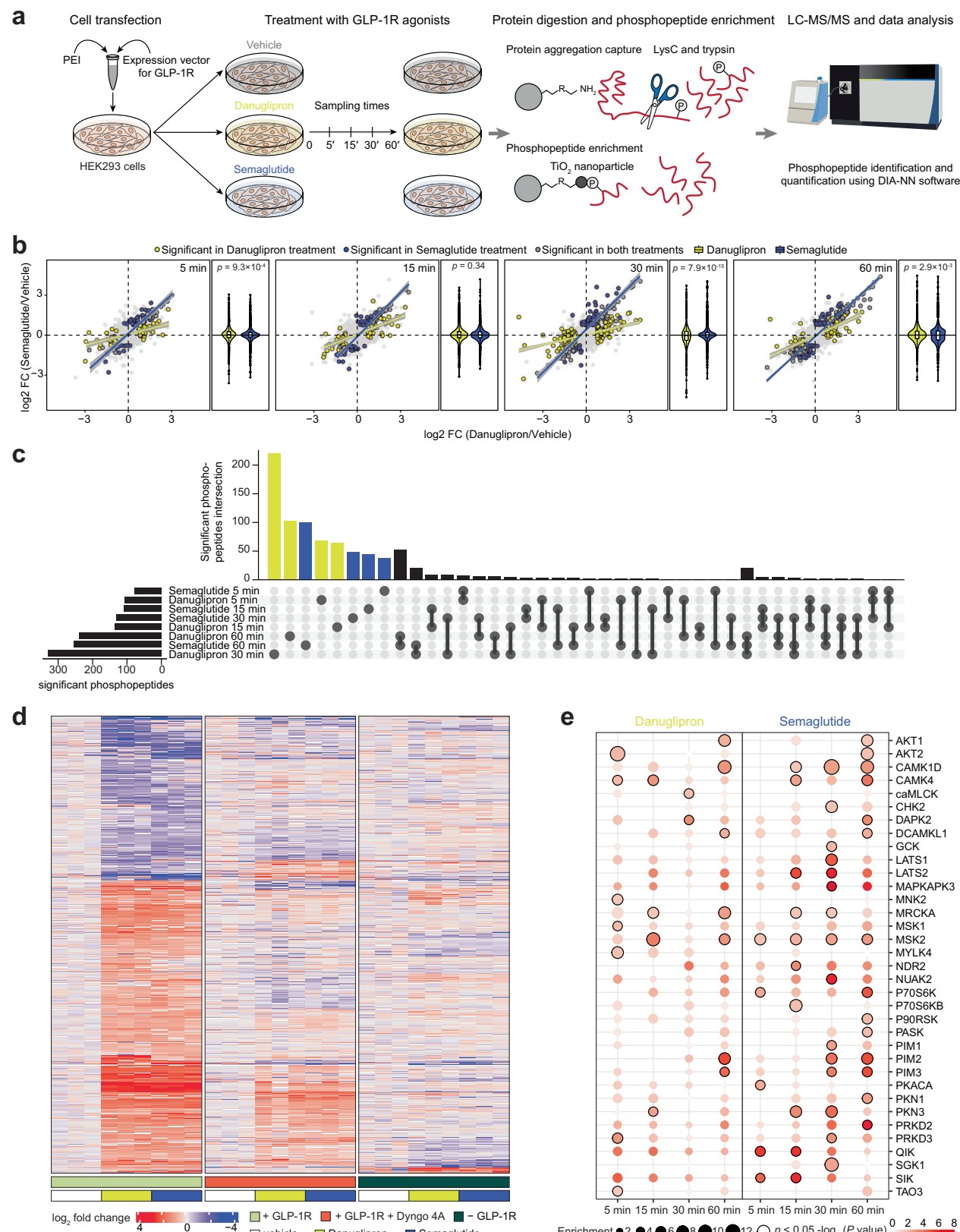

peptide treatments. Semaglutide resulted in trends that aligned with sustained signaling following receptor-mediated endocytosis, with maximal changes observed after 30–60 min. In contrast, danuglipron-treated showed distinctly different kinetics, demonstrating that different compounds elicit distinct compartment-specific response signatures.

**Clustering of GLP-1R agonist fingerprints correlates with ADRs**

The experiments described above provided a rationale for extending our investigation of transducer engagement to compartments beyond the plasma membrane. Using ebBRET, we extended our pathway profiling to endosomes, the Golgi apparatus and the endoplasmic reticulum. This endeavor led to the most thorough investigation into

**Fig. 5 | Changes in the phosphoproteome depend on functional selectivity and location bias. a** Phosphoproteomics workflow ($n = 3$ biologically independent samples). **b** 2-dimensional plots of mean log2-scaled fold change (FC) (time-matched) of the relative intensity of phosphopeptides in samples treated with danuglipron (x-axis) and semaglutide (y-axis) after 5, 15, 30 and 60 min of drug exposure (both compared to vehicle) with regression lines (grey bands represent 95% confidence interval) of proteins significantly regulated in each treatment. P values were calculated by a two-sided unpaired t-test. 2D plots are accompanied by violin plots of the mean log2 FC for both danuglipron and semaglutide at each time point (both compared to vehicle; internal box plots depict the 25th and 75th percentiles). Exact

p-values were calculated using the Kruskal-Wallis one way analysis of variance. **c** Upset plot showing shared significantly regulated phosphopeptides after treatment with either danuglipron or semaglutide compared to vehicle at each time point. **d** Heatmap of significant phosphopeptides (+GLP-1R) in control or GLP-1R-expressing cells treated with vehicle, danuglipron or semaglutide in the presence or absence of the internalization inhibitor Dyngo-4A. Experimental conditions are normalized to the corresponding vehicle and phosphopeptide changes are represented as log2 fold change. **e** Kinase substrate enrichment analysis[73] of the significantly regulated phosphopeptides for each drug and time point.

the signaling capacity of any GPCR to date (15 pathways in 4 compartments) (Fig. 6a, Supplementary Figs. 9–17 and Supplementary Data 8). From these, we extracted $E_{max}$ and $EC_{50}$ by applying stringent multiple testing correction ($P < 0.00625$ after Bonferroni correction). Exploration of these signaling neighborhoods revealed pharmacodynamic differences across pathways and compartments for the 8 tested drugs (Supplementary Fig. 18). Unsupervised hierarchical clustering of GLP-1R agonists based on these pharmacodynamic parameters resulted in the identification of several distinct drug clusters based on signaling profile similarities: (1) semaglutide; (2) lixisenatide; (3) liraglutide, GLP-1 (7-36) and exenatide and (4) danuglipron, glucagon and GLP-1 (1-37) (Fig. 6b and Supplementary Fig. 19). Examination of their cluster centers allowed us to determine the pathways, compartments and pharmacological parameters that most define the similarities within clusters and the differences across clusters. In short, positive k-means cluster centers were indicative of high potency or efficacy in a specific pathway and compartment, whereas negative k-means cluster centers could be interpreted as pathways that were underrepresented or not activated by a given drug cluster (Fig. 6c). Semaglutide was characterized by higher efficacy and potency in $G_{13}$ at early endosomes. In contrast, lixisenatide was characterized by higher potency in $G_{15}$ and $G_z$ at the endoplasmic reticulum and early endosomes, respectively. Efficacy in pathways related to $G_{i/o}$ signaling at the plasma membrane were underrepresented in these drug clusters. The cluster encompassing liraglutide and exenatide displayed higher efficacy for members of the $G_{i/o}$ family at the plasma membrane as well as increased GRK/Gβγ activity at the Golgi apparatus. Finally, the cluster, including danuglipron was largely defined by endoplasmic reticulum-localized recruitment of mGs and decreased recruitment of βARR at the plasma membrane.

We hypothesized that the differences in drug-induced signaling profiles might be related to ADR risk. We thus used the FDA adverse event reporting system (FAERS) to evaluate adverse event reports for all approved GLP-1R agonists included in our study: semaglutide, lixisenatide, liraglutide and exenatide (Fig. 6d). Our analysis revealed that more severe ADRs like pancreatitis were overrepresented in patients who received liraglutide and exenatide—a finding that aligns with clustering based on our comprehensive BRET profiling (Supplementary Figs. 20 and 21 and Supplementary Data 8, 9). In contrast, the similarity in amino acid sequence among peptide drugs was not associated with pancreatitis risk and was less predictive of ADRs than signaling neighborhoods (Supplementary Fig. 22). Determination of the k-means cluster centers revealed the cluster-specific molecular underpinnings to predict both protective and undesirable signaling signatures. Overall, these findings provide the molecular framework linking the differences in receptor conformation and signaling neighborhoods to clinical responses.

## Discussion

GPCRs are the most intensively studied of all drug targets and represent the largest target class of approved drugs and agents in clinical trials[42]. Despite our increased understanding of how these proteins operate and their recognized value for disease management, the drugs that target these receptors are not without their drawbacks, which

include lack of efficacy or safety issues. Moreover, the absence of scalable and sensitive tools to measure emerging paradigms like pathway selectivity and location bias—aspects of signaling that can contribute to ADRs—has made it difficult to integrate these properties into drug development programs.

The work presented here demonstrates how pluridimensional signaling can be measured by a singular robust, yet modular ebBRET-based biosensor platform, allowing for the characterization of pathway activation in an organelle-specific manner—an important aspect of GPCR biology that may become the new frontier in rational drug design[5,17,43,44]. Our data unequivocally show differences in the signaling fingerprints of GLP-1R agonists at the levels of transducer engagement, signaling kinetics and subcellular localization. However, the functional outcome of this pluridimensional signaling remains to be understood —especially in a native, tissue-specific context. Activation of GLP-1R in different tissues leads to insulin release, suppression of glucagon secretion, slowing of gastric emptying and increase in satiety[7]. It is therefore plausible that variation in the stoichiometry of the drug-specific GLP-1R transducerome across tissues could mediate differential outcomes and our work paves the way for future studies into such tissue-specific signaling profiles. Yet, coupling pathway-selective biosensors with compartment-specific markers, as we have done here, was sufficient to group GLP-1R agonists and correlate these groups with ADRs, suggesting that ADRs may, in part, be explained by pathway and compartment selectivity.

Moving forward, it will be important to extend the concept of signaling neighborhoods to primary human tissues to evaluate whether the profiles are conserved. Also, secretin receptor family crosstalk or interpatient variability are important parameters to consider in future work to fully comprehend the physiological impact of pathway-selective compartmentalized signaling. Another intriguing aspect of GLP-1R signaling that deserves attention is the potential for kinetic bias of drugs acting on this receptor, as this may modulate efficacy and potency[45,46]. Future research should therefore explore the consequences of different dosing regimens and pharmacokinetics of GLP-1R agonists in vivo on both signaling profiles and biological outcomes. Importantly, the firm establishment of causal relationships between transducer signatures and ADR risk requires future evaluations on a larger scale.

In addition to the plasma membrane, GPCR transducers like heterotrimeric G proteins, GRK and βARR can also be found in subcellular compartments. GPCR activation can dynamically affect the subcellular distribution and activity of these transducers, which may or may not be coupled with the trafficking properties of the receptor[14,15,17,19,20,47–51]. Using BRET and phosphoproteomics, we show that the signaling profiles of peptide and small molecule agonists of GLP-1R are rather different, which is consistent with our observation that they engage distinct residue contact networks. Our analyses reveal that GLP-1R agonists can elicit drastically different downstream signaling profiles. For instance, we show that semaglutide and lixisenatide had a lower efficacy for $G_{i/o}$ than liraglutide and exenatide specifically at the plasma membrane and coincidentally have a lower incidence of pancreatitis. In addition to patterns specific to individual compounds, drug class-specific differences were observed. Specifically, βARR recruitment at

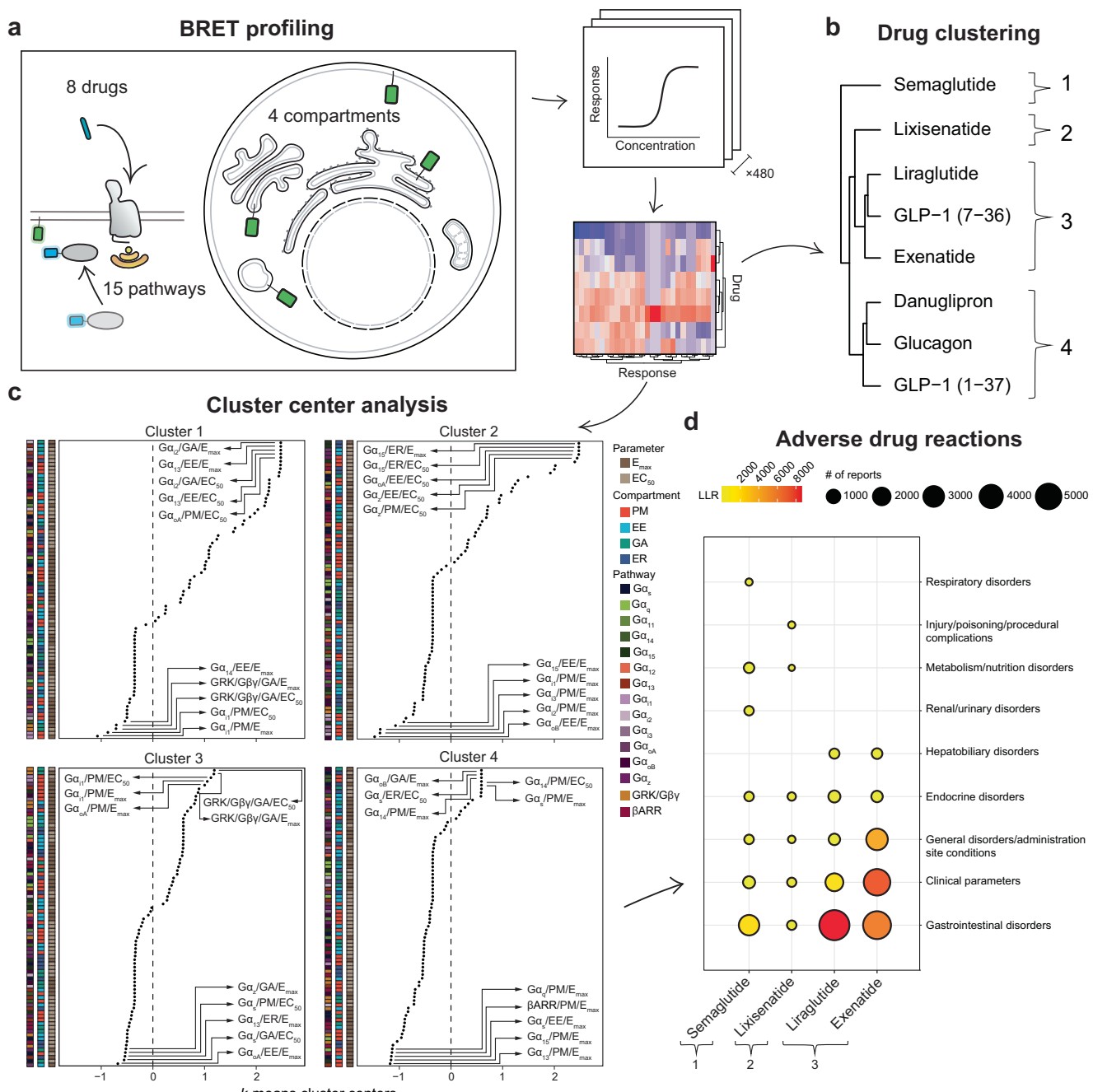

**Fig. 6 | Hierarchical clustering of GLP-1R agonist fingerprints correlates with ADRs. a** Schematic of the experimental workflow for analyzing the responses of 8 GLP-1R agonists in 15 pathways and 4 cellular compartments by ebBRET. Efficacy and potency were extracted from concentration-response curves and aggregated before $k$-means clustering. **b** Hierarchical clustering of GLP-1R agonists based on signaling neighborhood profiles. **c** Cluster centers of distinct drug clusters identified by $k$-means clustering. Defining cluster features are listed by rank order, and the top and bottom five features are highlighted for each cluster. Positive $k$-means cluster centers reflected compartmentalized pathway engagement with high potency or efficacy, while negative $k$-means cluster centers referred to pathways that were not activated or less activated by a given drug cluster. **d** Significant adverse reactions (ADRs) have been estimated from reports submitted to the FDA Adverse Event Reporting database (FAERS) by transformation into a log likelihood ratio (LLR). All 48 unique low-level ADRs are mapped to the Medical Dictionary for Regulatory Activities (MedDRA), and aggregated "System Organ Class" (SOC) ontology levels.

the plasma membrane and receptor endocytosis, which have been linked to nausea[29], were revealed to be common defining features of all incretin mimetics, but not the small molecule drug danuglipron. However, the impact of danuglipron-induced GLP-1R activation at the Golgi and endoplasmic reticulum, but not at endosomes, coupled with extensive changes to the cellular phosphoproteome remains unknown. This information has enormous implications for drug development and clinical practice as oral small molecule compounds

that target Class B GPCRs are increasingly sought after due to improved adherence[8, 9, 11].

In the present study, we show that location- and pathway-selective signaling profiles correlate with compound safety. These results are consistent with recent findings that ADRs correlated better with drug signaling profiles than with pharmacokinetic properties[52]. Using a different approach, our analyses reveal that sequence modifications that prevent degradation or increase protein binding had a greater effect

on defining drug fingerprints than their homology to the human endogenous peptide. Specifically, exenatide, which is derived from a peptide found in the venom of the Gila monster that is separated from human biology by around 320 million years of evolution (~50% sequence identity), closely resembles transducer signatures and safety profiles of liraglutide—a modified human incretin. This demonstrates that, for the subset of compounds tested in this study, GPCR signaling neighborhoods serve as a better predictor of clinical outcomes than the structural homology of compounds. Future conscious efforts aimed at shaping these signaling neighborhoods through data-driven drug development may increase our chances of minimizing adverse events.

## Methods

### Reagents
D-PBS, DMEM, Trypsin, PBS, penicillin/streptomycin, fetal bovine serum (FBS) were from Gibco (Thermo Fisher Scientific, Waltham, MA, USA). Polyethylenimine (PEI) was purchased from Alfa Aesar (Thermo Fisher Scientific, Waltham, MA, USA). Glucagon was from Sigma Aldrich (Saint Louis, MO, USA). GLP-1 (1-37), GLP-1 (7-36), exenatide, liraglutide, semaglutide, lixisenatide and danuglipron were from either BACHEM (Bubendorf, Switzerland) or MedChemExpress LLC (Monmouth Junction, NJ, USA). Dyngo-4A was from ApexBio (Houston, TX, USA). Coelenterazine 400a was purchased from Nanolight Technologies (Pinetop, AZ, USA).

### Plasmid DNA constructs
GLP-1R-$R$luc8[53], p63RhoGEF-$R$lucII, Rap1Gap-$R$lucII, PDZRhoGEF-$R$lucII[5], GRK2-D110A-$R$lucII[54], β-arrestin-2-$R$lucII[55], rGFP-CAAX and rGFP-FYVE[43] and tdrGFP-Giantin and tdrGFP-PTP1B[17] have been described previously. Human GLP-1R was generously given by Domain Therapeutics NA Inc. SNAP-GLP-1R was from Novo Nordisk. GLP-1R-$R$lucII was generated by amplifying human GLP-1R from Domain Therapeutics NA Inc. with flanking restriction sites for NheI and BamHI followed by subcloning into pcDNA3.1-(GFP[10])-$R$lucII to replace GFP[10] with GLP-1R. Human Gα$_{i1}$, Gα$_{i2}$, Gα$_{i3}$, Gα$_{oA}$, Gα$_{oB}$, Gα$_z$, Gα$_q$, Gα$_{11}$, Gα$_{14}$, Gα$_{15}$, Gα$_{12}$ and Gα$_{13}$ were purchased from cDNA.org (Bloomsburg University, Bloomsburg, PA). $R$luc8-mGs and mRuby2-mGs were generated by subcloning from NES-Venus-mGs[31] into $R$luc8-C1 and mRuby2-C1 vectors using EcoRI and XhoI. All plasmid constructs were verified by Sanger sequencing.

### Cell culture and transfection
HEK293 cells were cultured in DMEM supplemented with 10% FBS, 1% penicillin/streptomycin, propagated in plastic flasks and grown at 37 °C in 5% $CO_2$ and 90% humidity. Cells (350,000 in 1 ml) were transfected in suspension with 1.0 μg of plasmid DNA complexed with linear polyethyleneimine (PEI; MW 25,000, 3:1 PEI:DNA ratio) and seeded (3.5 x $10^4$ cells/well) in white 96-well plates. All cell lines were regularly tested for mycoplasma contamination.

### BRET assays
**ebBRET.** To monitor the recruitment of transducers, HEK293 cells were transfected with GLP-1R, $R$luc8-mGs, GRK2-D110A-$R$lucII, or β-arrestin-2-$R$lucII and rGFP-CAAX, rGFP-FYVE, tdrGFP-Giantin or tdrGFP-PTP1B. To monitor the recruitment of effectors to transducers (GEMTA), HEK293 cells were transfected with GLP-1R, Gα, p63RhoGEF-$R$lucII, Rap1Gap-$R$lucII or PDZRhoGEF-$R$lucII and rGFP-CAAX, rGFP-FYVE, tdrGFP-Giantin or tdrGFP-PTP1B. Receptor trafficking was measured in HEK293 cells that had been transfected with GLP-1R-$R$luc and rGFP-CAAX, rGFP-FYVE, tdrGFP-Giantin or tdrGFP-PTP1B. After a 48-hour incubation, cells were washed once with HBSS and maintained in the same buffer. Prior to BRET measurements, cells were stimulated with agonist and incubated with coelenterazine 400a (2.5 μM; 5 min). See Supplementary Data 10 for more information.

**BRET measurements.** Plates were read on a Tecan Spark multimode microplate reader (Männedorf, Switzerland) equipped with the following filters for BRET[2]: 400/70 nm (donor) and 515/20 nm (acceptor) for detecting the $R$lucII/$R$luc8 (donor) and rGFP (acceptor) light emissions, respectively.

**Polar area diagrams.** Best fit values were excluded if $EC_{50}$ was outside of the concentration range tested or if the span was negative. One-sided extra sum-of-squares F test was employed to test whether top parameter of the non-linear regression was statistically distinguishable from the bottom. The Bonferroni correction was applied to account for multiple comparisons across the eight drugs in each pathway ($P < 0.00625$). All further data processing was performed in R (version 4.2.1).

### Confocal microscopy
**mGs translocation.** HEK293 cells were transfected in suspension with GLP-1R, mRuby2-mGs, and rGFP-CAAX, rGFP-FYVE or tdrGFP-Giantin and grown in four compartment 35 mm glass bottom dishes. After 48 h, the cells were washed with HBSS before adding 1 μM semaglutide. Cells were imaged every 15 s for 30 min. Confocal images were acquired using a Zeiss 980 laser scanning microscope. Images were acquired using a 40×, 1.2 NA C-Apochromat objective and the 488 nm and the 594 nm laser lines were used to excite rGFP and mRuby2 fluorophores, respectively. To quantify colocalization between mRuby2-mGs and tdrGFP-Giantin, green particles were selected as regions of interest and the mean intensity of red fluorescence per area was measured throughout the duration of the timelapse in ImageJ2 2.9.0.

**GLP-1R trafficking.** HEK293 cells seeded on 8 well μ-Slide (ibidi) were transfected with SNAP-GLP-1R and compartment markers fused to rGFP. The following day, cells were labeled with 5 μM SNAP-Surface® Alexa Fluor® 647 (New England Biolabs) for 15 min at 37 °C. Cells were then washed and exposed to 1 μM semaglutide for the following incubation times: 10 min for PM (rGFP-CAAX), 20 min for EE (rGFP-FYVE) and 30 min for GA (tdrGFP-Giantin). Finally, cells were fixed with PFA 2% before adding ibidi Mounting Medium with DAPI. Confocal microscopy was carried out using an SP8 LIGHTNING confocal microscope (Leica Microsystems) with a 63× oil immersion lens, [410-483 nm, 493-560 nm and 710-775 nm for DAPI, rGFP and Alexa Fluor 647 respectively]. Images were treated using ImageJ2 2.9.0 software.

### Adverse drug event analysis
Post-marketing adverse events for all approved and available GLP-1R-targeting agents included in our study were obtained. Significant adverse reactions (ADRs) have been estimated from reports submitted to the FDA Adverse Event Reporting database (FAERS) by healthcare professionals by transformation into a log likelihood ratio (LLR)[56] as provided by Open Targets (https://platform.opentargets.org/)[57] and which corrects for the prevalence of a given drug and the frequency of an event across drugs. In total, 48 unique ADRs were retrieved and mapped to the Medical Dictionary for Regulatory Activities (MedDRA), which was further aggregated and summed into 9 "System Organ Class" (SOC) ontology-levels.

### Residue contacts analysis
Six GLP-1R structures were selected from the Protein Data Bank, four in active and two in inactive conformations, respectively. The active structures are in complex with: GLP-1 (7-36) [PDB:6X18[11]], semaglutide [PDB:7KI0[58]], exenatide [PDB:7LLL[59]] and non-peptide agonist PF-06882961/danuglipron [PDB:6X1A[11]]. The two inactive structures in complex with two negative allosteric modulators, respectively PF-06372222 [PDB: 6KJV[60]] and NNC0640 [PDB: 5VEX[61]]. Structures in complex with a signaling protein were set as the reference structures

for the active state (100%-degree activation) and structures with a highly closed conformation were set as the reference structures for the inactive state (0%-degree activation) based on a maximum measured distance between 2x46 to 6x37. The Cα atom distance pairs for each structure were compared to the reference structures and the mean distance to the active structures and the mean distance to the inactive structures were calculated. These distances to the reference structure sets were then converted into an "activation score" by subtracting the mean distance to the inactive-state structures from the mean distance to the active-state structures. The activation score was converted into a percentage activation based on the minimum and maximum activation scores for all structures in that class (https://docs.gpcrdb.org/structures.html?highlight=activation#structure-descriptors). In our analysis, all four selected active structures displayed 100% normalized degree activation. Conversely, the two reference inactive state structures both displayed 0%-degree activation. See Supplementary Data 1 for more information.

Both the analyses of the receptor contact pairs and the interaction types were performed using the GPCRdb webserver for comparative structure analysis (see https://review.gpcrdb.org/structure_comparison/comparative_analysis)[3, 62]. Briefly, contact analysis was performed considering sidechain-sidechain and sidechain-backbone interactions among inter-segment residues and only sidechain-sidechain interactions among intra-segment contacts (as the GPCRdb webserver default settings). For each receptor residue, non-hydrogen atoms in close proximity of non-hydrogen atoms from neighboring residues are taken into account. The potential contacts are further evaluated based on atom and residue types and their distance. For each of the contact types, the default maximum distances are ionic (4.5 Å), polar (4 Å), aromatic (stacking 5.5 Å and cation-π 6.6 Å), hydrophobic (4.5 Å) and Van der Waals contacts (1.1 times their combined VdW radii).

Corresponding residue positions in the structures were identified with the structure-based GPCRdb generic residue numbering system that is based on the structure-based generic residue numbering system for class B1 (Wootten numbering). This avoids gaps and mismatching after structural alignments of receptors due to missing residues and/or different helix bulges/constrictions among the structures.

Following this numbering system, not all residues can be assigned to generic numbers; in fact, only the TM helices, H8 and structurally conserved loop segments with annotated generic residue numbers were included, excluding the extracellular domain. The included structures varied in their coverage from 263 (5VEX) to 390 (6X1A) resolved residues (57% and 84% of the total GLP-1R length of 463 residues). We hence restricted the residue contact analysis to only the shared 219 residues among all 6 structures.

The obtained shared contact pairs were represented in a Venn-like diagram (*Supervenn*) where each row is a set of PDB-specific contact pairs, and the overlapping parts (groupings) correspond to intersections of sets (i.e. shared residue-pairs among sets). The sizes of sets and their groupings are proportional, but the order of elements is not preserved. A combinatorial optimization algorithm was applied that rearranges the groupings (the columns of the array plotted) by occurrence (groupings that are in more sets go first). The columns on the right represent, for each PDB, respectively: the set sizes (N. contact pairs), the residues included in the analysis (N. total residues), and the unshared residues excluded from the analysis (N. excluded not shared residues). The numbers on the top (Number of overlapping structures) show how many sets fit into this intersection (i.e. how many PDBs show that particular grouping of contact pairs). The numbers below (number of total contact pairs) represent a scale to give indications of the proportion of the number of residue contacts.

All structural representations were produced using UCSF Chimera[63]. Contacts are shown by the alpha carbon of the interacting residue pair (showed in ball and stick representation).

## Phosphoproteomics

HEK293 cells (350,000 cells/mL) were transfected in suspension using 3 μg of linear PEI (MW 25,000) per 1 μg of plasmid DNA and seeded onto poly-D-lysine-coated 60 mm dishes at a density of 1,750,000 cells per dish. After a 24-h incubation, cells were washed twice with PBS and placed in starvation media (DMEM with penicillin/streptomycin). After a 48-h incubation, cells were washed twice with HBSS and maintained in the same buffer. For timecourse experiments, cells were stimulated for 0, 5, 15, 30 and 60 min with vehicle (1% DMSO), semaglutide or danuglipron (1 μM). For probing the internalization-dependent changes to the phosphoproteome, cells were preincubated with Dyngo-4A (50 μM) for 30 min prior to stimulation for 30 min with vehicle (1% DMSO), semaglutide or danuglipron (1 μM). Cell extracts were prepared as previously described[64–66]. In short, 200 μL of the lysis buffer preheated at 95 °C (4% sodium dodecyl sulfate in 50 mM Tris adjusted to pH 8.5) was added to the cells. The samples were then denatured at 95 °C for 10 min, placed on ice and then sonicated using a probe sonicator (3 sec on, 3 sec off pulse, 1 min, 30% amplitude) (Branson). Lysate protein concentration was determined using the Pierce bicinchoninic acid assay kit (Thermo Fisher Scientific) according to the manufacturer's instructions. The lysates were reduced using 5 mM dithiothreitol for 1 h at room temperature and alkylated using 15 mM iodoacetamide for 1 h at room temperature in the dark. Proteins were precipitated and digested on beads using MagReSyn Amine magnetic beads (ReSyn Biosciences) according to the manufacturer's instructions. Protein digestion was conducted overnight at 37 °C with lysyl endopeptidase (LysC, Wako) (1:150 LysC/protein, w/w) and trypsin (1:300 trypsin/protein, w/w). 20% TFA, 2 M glycolic acid in 80% ACN were added to the samples at a 1:1 ratio for phosphopeptide enrichment using MagReSyn TiO₂ magnetic beads (ReSyn Biosciences) according to the manufacturer's instructions. The enriched phosphopeptides were dried overnight in a SpeedVac, resuspended into 5% formic acid (FA), desalted using StageTip (Thermo Fisher Scientific) according to the manufacturer's instructions and dried in a SpeedVac. The experiment was performed in biological triplicates.

## Mass spectrometry analysis

All parameters and settings regarding the acquisition of the proteomics data can be found in Supplementary Data 11, 12. For data acquisition, the instrument was set to peptide mode, with a trapping pressure set as low (2mTorr). An EASY-Spray Source (Thermo Fisher Scientific) was used and the column temperature was maintained with the integrated, temperature-controlled heater at 55 °C throughout the experiment. For the data presented in Fig. 5d, the column was heated at 50 °C following the manufacturer's instructions (IonOpticks).

## Bioinformatics and data analysis for proteomics

Acquired raw files were converted to mzML format by MSConvert (version 3.0.21258)[67] applying peak picking of the mass spectra with the vendor-provided algorithm (Thermo Fisher Scientific). The mzML files were analyzed using the FragPipe GUI v18 for both library and DIA search. MSFragger[68] was used to search files against the human Swissprot database (20,409 entries). Trypsin with up to 2 missed cleavages was set as digestion enzyme, oxidation of methionine, acetylation of the N-terminus and phosphorylation on serine, threonine and tyrosine (maximal occurrence was set to three) were set as variable modifications. Carbamidomethylation of cysteine residues was set as a fixed modification. Peptide length was restricted to 7 to 50 amino acids, and 500 to 5000 Da. The precursor tolerance was set to ±20 ppm, fragment mass tolerance to 20 ppm and MSBooster was enabled. The resulting peptide-spectrum matches were adjusted to a 1% false discovery rate with Percolator[69] as part of the Philosopher toolkit (v4.4)[70] and converted to a spectra library. DIA files were analyzed by DIA-NN 1.8.1[71, 72] using the previously generated library and monitor-command

mode. Peptide quantification was performed based on the fragment elution profiles from the MS2 level (area under the curve).

All further data processing was performed in R (version 4.2.1). Only phosphopeptides with a localization score of at least 0.75 were considered for further analysis. Peptides quantified with different charge states were aggregated and only phosphopeptides with at least 2 quantifications for each time point, treatment and replicate were kept. Intensities were normalized by median centering and log2-scaling. Missing values were imputed by $k$-NN imputation. All statistical comparisons were performed based on homoscedastic two-tailed Student's $t$-tests. For the kinase substrate enrichment analysis, the Kinase Library online analytical tool[73] was used with predetermined foreground and background sets based on previously determined significance thresholds ($P < 0.05$; log2 fold change >1). All remaining settings were left as default.

### Data analysis

Data analysis was performed using GraphPad Prism 9.4.0 (GraphPad Software, San Diego, CA, USA). Quantitative data are expressed as the mean and error bars represent the standard error of the mean (SEM) unless otherwise indicated. Raw BRET was calculated by dividing the emission intensity at 515/20 nm by the emission intensity at 400/70 nm. netBRET was determined by subtracting the ratio measured from cells expressing only the BRET donor from cells expressing the BRET donor and acceptor. ΔBRET was calculated by subtracting the BRET ratio of vehicle-treated cells from the BRET ratio of agonist-treated cells. Curve fitting was performed by three or four parameter nonlinear regression. Statistical analyses were performed using one-sided extra sum-of-squares F test for comparison of fit parameters with Bonferroni correction. Potency and efficacy were extracted from drug-directed pathway- and compartment-selective activity and represented as a heatmap. For potency, pathway and compartment combinations required at least one measured value and missing values were inputted with $10^{-2}$ M. All data for potency and efficacy were $z$-scaled (center = 0, standard deviation = 1). Combined data sets were then clustered by hierarchical clustering (using the Manhattan distance and ward.D2 clustering) for both drugs and parameters. Drug clusters were determined by $k$-means clustering. Cluster membership probability was assessed by fuzzy clustering[74]. The predictiveness of ADRs was based on the dendrogram of sequence homology or signaling neighborhoods compared to adverse event risk from the FDA Adverse Event Reporting database. Specifically, entanglement score was calculated as the L-norm distance between two dendrograms with a score of 1 indicating perfect alignment and a score of 0 indicating an absence of correlation.

### Reporting summary

Further information on research design is available in the Nature Portfolio Reporting Summary linked to this article.

## Data availability

All data needed to evaluate the conclusions in the paper are present in the Source Data file. mG probes are available from N.A.L. upon request. Some of the biosensors used in present study are protected by patents, but all are available for academic research under regular material transfer agreement (MTA) upon request to M.B. The mass spectrometry proteomics data files have been deposited to ProteomeXchange Consortium via the PRIDE partner repository with the data identifier PXD037472. All other study data are included in the article and/or supporting information. Source data are provided with this paper.

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

## Acknowledgements

S.C.W. is supported by the Swedish Society for Medical Research (P18-0098; PD20-0153). S.K. acknowledges funding from the Onassis Foundation (F ZR 072-1/ 2021-2022). P.S. is funded by the Swedish Research Council (2022-00323). N.A.L. is supported by National Institutes of Health grants GM130142 and GM145284. A.S.H. is supported by the Lundbeck Foundation (R278-2018-180). M.B. is funded by the CIHR (FDN-148431) and holds a Canada Research Chair in Signal Transduction and Molecular Pharmacology. V.M.L. is supported by the Swedish Research Council (grant agreement numbers 2019-01837 and 2021-02801), by the EU/EFPIA/OICR/McGill/KTH/Diamond Innovative Medicines Initiative 2 Joint Undertaking (EUbOPEN grant number 875510), by the Swedish Strategic Research Programme in Diabetes (SFO Diabetes) and Stem Cells and Regenerative Medicine (SFO StratRegen), by the European Union's Horizon 2020 research and innovation program U-PGx (grant agreement number 668353), and by the Robert Bosch Foundation, Stuttgart, Germany.

## Author contributions

M.B., V.M.L and S.C.W. designed the study. A.M., S.K., P.S., C.M.B., A.B., E.B.-T., K.M., J.V.O., R.A.Z., A.S.H., M.B., V.M.L. and S.C.W. analyzed and interpreted the data. A.M., S.K., P.S., C.M.B., E.B.-T., K.M., I.P., D-C.S. and S.C.W. performed experiments presented in this study. K.M., C.L.G. and N.A.L. generated and characterized DNA constructs used in this study. A.M., P.S., C.M.B., A.B., E.B.-T., A.S.H., M.B., V.M.L. and S.C.W. prepared figures for publication. M.B., V.M.L and S.C.W. wrote the manuscript with input from all authors.

## Funding

## Competing interests

M.B. is the president of the scientific advisory board for Domain Therapeutics. M.B. and C.L.G. have filed patent application (US20200256869A1) related to some of the biosensors used in this work and the technology has been licensed to Domain Therapeutics. All biosensors are available for academic research through regular material transfer agreement. V.M.L. is co-founder, the CEO, and shareholder of HepaPredict AB, as well as co-founder and shareholder of PersoMedix AB. A.M., S.K., P.S., C.M.B., A.B., E.B.-T., K.M., I.P., D-C.S., N.A.L., J.V.O., R.A.Z., A.S.H. and S.C.W. declare no competing interests.
