## [Peer Review File · Nature Communications]

REVIEWER COMMENTS

Reviewer #1 (Remarks to the Author):

The ability of GPCRs to signal from different subcellular compartments is now well-established and has been demonstrated for multiple GPCRs. In this manuscript the authors aimed to characterize the signaling properties of GLP-1R at different subcellular locations and in relation to possible adverse side-effects of the drugs. I think the data is compelling and the overall layout is comprehensive and thorough and could be a suitable template for characterization of other receptors in the future. I have several questions and concerns which I am requesting the authors to address in their revised manuscript.

1- Some of the structures used in the analysis in figure 1 are truncated or with engineered stabilizing mutations. Also, some structures were acquired with G proteins coupled with the receptor. Do the authors anticipate this could affect some of the contact maps and downstream analysis or have they taken this into consideration?

2- Figure 1 is a bit hard to read. For example, what is the last column in figure 1a? What is the lower x-axis in the figure 1a? What are the colors in figure 1a represent? Having this information explicitly mentioned in the figure or in the caption would be very helpful.

3- Generally, the methods section needs to be updated to include detailed practical information about the measurements. For example, the cell culture media condition, FBS%, amount of DNA used for transfection, duration of expression, concentration of coelenterazine, the buffer that BRET measurement was done in, how is BRET value calculated from the raw data in the plots, if there are any corrections to the raw data, etc should be included or clearly referenced. With the current level of details, it is hard for other labs to try to replicate the data.

Regarding the phosphoproteomic analysis:

Overall, this section seemed a bit disconnected from the rest of the manuscript. I suggest the authors incorporate it with the rest of the manuscript and discussion in a more coherent way.

4- Does the protein extraction protocol used in the assay recover proteins from internal organelles?
Since

5- The result that shows enrichment of very different sets of phosphorylated proteins in the presence of different agonists and at different time points is interesting. However, an important control would be performing the experiment with cells not expressing the receptor but treated with the compounds. Have the authors done such a measurement?

6- From the methods it appears that a phosphatase inhibitor was not used during sample preparation. Usually this is added in such phosphoproteomic analysis to avoid unwanted loss of phosphorylation.

7- Can the authors comment on how many phosphorylations are on each peptide? In the bioinformatics methods they mention that they allow up to 3 co-occurring phosphorylations for a single peptide in their search parameters? TiO₂ will enrich heavily phosphorylated peptides with greater affinity, so if there is one peptide that maybe occurs in a mono- and di-phospho state did the authors consider that in abundance calculations.

8- In the mass spectrometry methods, I'm assuming these parameters are for an MS² experiment: "30,000 resolution with 27 % NCE, an AGC target was set to 1000%". The NCE is very high and phosphorylations can be ejected during HCD. Have authors verified that this is a safe range of energy

for this analysis?

8- When the authors did comparative quantification between samples, are they comparing the abundance of a specific peptide, modification state, and charge state? The charge and modification state will change the ionization efficiency a great deal and this will impact the calculated abundance calculation. For example, are they comparing some peptide like N-GER(pT)SIP-C in the 2+ charge state directly between different conditions?

9- Also, is the abundance of a peptide defined as number of scans? Or the actual area under the curve of ion current for a specific peptide on the LC timescale?

10- The written mass spectrometry methods are pretty well-written. I assume the Lumos was set into peptide mode and the trapping pressure was "low" (2 mTorr?). Please specify that parameter as well. Please also specify the source used on the front end of the mass spec, e.g. is it a HESI, NanoFlex, etc.?

Reviewer #2 (Remarks to the Author):

Authors

GPCRs classically signal from the plasma membrane, but several studies indicate that they can maintain signaling from intracellular compartments, largely in the endosomal pathway. So that authors ask whether membrane permeable small molecule agonists of the GLP1 receptor have different cellular signaling effects than do peptide agonists, which do not enter cells. Not surprisingly the answer is 'yes' as is established by the authors in studies carried out using an array of techniques in HEK293 cells. In addition, the small molecule agent under study (danuglipron) was shown to be less biased toward beta-arrestin mediated signaling than peptide agents. The authors extrapolate from their HEK293 results to some conclusions regarding likelihood of adverse drug reactions of the agents. There are a number of issues that need author attention:

--- The gist of this paper is that (from the authors' Discussion) "Stimulation of GLP-1R with peptide agonists and cell permeable small molecule agonists results in different subcellular activation." This is perhaps not surprising. I think that it is problematic to extrapolate, however, from studies in HEK293 cells with overexpressed signaling proteins to predictions about responses and adverse drug reactions in human patients. So, although the data presented are of interest, the authors need to guard against overinterpretation.

--- We are entering into an enhanced data sharing era, which requires comprehensive data sharing of all data types. In proteomics, data sharing has been the norm for more than 10 years. In general, detailed sequence data is shared for all protein mass spectrometry data, usually via one of the members of the ProteomeXchange Consortium. In Europe, this would be PRIDE at <https://www.ebi.ac.uk/pride/>. Upon deposition, PRIDE provides a link and reviewer login credentials to allow review of the data. The authors have indicated they they have so deposited the data, but inserting the given number gave a message, "Sorry, no projects found for search term PXD037472 and the current set of active filters." More significantly, the data have not been provided in curated form, e.g. in a spreadsheet, reporting the abundance every phosphopeptide (or every phospho-site) in every sample, along with calculations used to create Figure 5. The phosphoproteomic data sound interesting, but without access to the data, it is impossible to assess data quality or to replicate the basic calculations.

----Focusing on the phosphoproteomics some information was missing from the Methods section. The phospho-proteomic studies seem to confirm the conclusion that the small molecule agent used produces different signaling responses than one of the peptide agonists. Nevertheless, the presentation of phosphoproteomic data was superficial and did not even tell us any of the phosphorylation sites that were changed in response to the agents.

--- The proteomics methods state "The experiment was performed in biological triplicates." That means that there were 45 total samples... 5 time points for each of three treatments in triplicate. 5 x 3 x 3. Please confirm. Having a spreadsheet with all phosphopeptides in all samples would have clarified this.

--- Authors should be more explicit about phosphopeptide quantification. Apparently they used a label-free method, but the principle of the quantification is vague. Presumably, the method calculated the area under the MS1 curve to get integrated intensity values. Please provide this information. For quality control, it would be helpful to show concordance between two peptides with the same phospho-site (e.g. from different charge states or missed cleavages). Availability of a spreadsheet with the intensities for each peptide in each sample would have been helpful. It was not among the Supplementary Files.

--- It is a disappointing that a labeling method was not employed such as TMT. Since this technique directly compares peptide abundances in all samples, the data would have been stronger and more phosphopeptides could have been reported. Matching peptide quantification from different MS runs is always problematic.

--- In the phosphoproteomic data, it would be helpful to know if any of the agonists increased phosphorylation of the COOH-tail of the GLP1 receptor, particularly at sites thought to be targeted by GRKs.

--- Figure 5d should give the specific phosphorylation site and the phosphopeptide sequences listed in the legend for transparency.

--- Figure 5e is obscure. What would be more helpful would be a report of the sequence motifs that would be formed from differentially phosphorylated sites. These motifs can then be mapped to kinases using data from a recent paper published in Nature (Nature 613, 759–766 (2023). DOI: 10.1038/s41586-022-05575-3). There are a number of programs that can generate the associated kinase target sequence logos such as PTM-Logo (<https://hpcwebapps.cit.nih.gov/PTMLogo/>). The black box nature of the KSEA analysis undermines confidence in the conclusions.

--- The authors pick out the Ephrin type-B receptor 4 and the AMP-dependent kinase as two protein kinases that are involved in ER stress. Regulation of the ER stress/UFR mechanism is not the primary role of these kinases however and the implication in the paper that Danuglipron impairs ER function is not well supported.

--- The plots shown in Figure 5b are a bit problematic. According to the diagram in Figure 5a, the same vehicle sample was used for calculation of both the y axis value ($\log_2(\text{Semaglutide}/\text{Vehicle})$) and the x-axis value $\log_2(\text{Danuglipron}/\text{Vehicle})$. As a consequence, the nice correlations shown can be called into question since the same term (vehicle) is used in both x and y axis. However, the main conclusion that the responses differ between the two agents is not in question, so this is not a crucial point, although worth clarifying in the paper. In addition, it is not clear from the presentation that independent time control vehicle samples were collected for each time point. Data sharing would have helped sort this out.

--- It seems questionable whether studies with overexpressed GLP1-R and effectors in HEK293 cells successfully mimic physiological responses with signaling proteins at endogenous levels. The type of study described in the paper can tell us if a given protein-protein interaction CAN occur and whether a drug CAN effect a given response, but does not establish whether these responses DO occur in native tissue. Authors need to address this limitation and consider animal studies that may strengthen the conclusions. (This is a problem with the whole promiscuous GPCR-G protein coupling story.)

--- The abstract is too generic. It sounds like it was written with ChatGPT or similar. "...translating signaling neighborhoods into functionally distinct cellular outcomes and clinical consequences." is not going to be understood by many. It seems like the abstract should say that cell permeant small molecule agonists have different cellular effects than do peptidic agents that cannot enter the cell. This is more clearly stated in the Discussion in the section, "Stimulation of GLP-1R with peptide and small molecule agonists results in selective subcellular activation."

--- The statement, "To our surprise, we also observed a convergence and accumulation of mGs positive vesicles to the Golgi apparatus suggesting that GLP-1R may engage in endosome-to-Golgi retrograde transport..." should be reconsidered. Every plasma membrane integral membrane protein is translated on the rough ER and must advance through the Golgi and through secretory vesicles to

get to the plasma membrane. So the presence of signal emanating from any of those compartments does not imply retrograde transport.

--- Extrapolation from results with overexpressed GLP1-R and effectors in HEK293 cells to likelihood of adverse effects in human patients seems risky. Animal studies would seem more useful. It is advised to remove the whole ADR element from this paper or to mention it only briefly. Its inclusion reduces confidence in the main conclusions of this paper.

--- The disclosure is incomplete. It needs to state whether Domain Therapeutics is working on agents that target GLP1-R. This is unclear from the DT web site.

--- Minor: There seems to be a lot of pseudo-systems-biology jargon like the term 'rewiring' which is used in so many ways that it is not clear what is meant in this paper.

Reviewer #3 (Remarks to the Author):

This is a very well-performed manuscript that characterizes drug efficacy and location bias across a range of agonists at the GLP-1R, a very important drug target in diabetes and obesity (probably the most important drug target). The experiments are rigorously performed by groups that have developed the enhanced bystander BRET (ebBRET) platform that allows an assessment of pathway activation in a location-specific manner. The experiments are well-performed and the number of assays performed very impressive (especially looking across 15 assays at 4 locations and 8 conditions). The overall conclusion (that these drugs have wildly different patterns of signaling, especially based on location) is supported by the data. But the weakness of the manuscript are the conclusions/claims that are not fully supported by the data (such as the adverse drug reactions) and data that really is not central to the major claims and is sort of supportive but not fleshed out well (phosphoproteomics). So I think that the manuscript largely requires a toning down of claims and some different types of analyses.

Major Comments:

- Figure 1 – The comparative structure analysis is nice, but doesn't add much unless those insights are used to guide experiments. I think this may be better served as a supplement as no real conclusions arise from it other than that they have different binding modes (not surprising).

- Figure 2 – This is an issue with Figure 6 as well, but I don't think the comparisons across EC50 in (d) are not insightful and, if anything, misleading. Those similarities are primarily related to affinity and not efficacy, and since affinity varies considerably between different peptides and small molecule agonists, the obtained results do not provide meaningful insights. While the Jaccard index provides some quantitative assessment of similarity, an assessment of bias would be better performed using the approaches recently outlined in IUPHAR community guidelines (<https://pubmed.ncbi.nlm.nih.gov/35106752/>).

- Figure 3 – From the text and the figure it is not clear to me whether the Gs is only noted at the Golgi. Please show merged images to better demonstrate colocalization of the location markers and mini Gs constructs.

- Figure 4 – The time course of the signal in the Golgi would be interesting to note. Is there signal in the Golgi before 30 minutes? It is important to differentiate the retromer based process of endosome to Golgi as opposed to what has been seen at the beta 1 adrenergic and adenosine A1 receptors where there are distinct pools of Golgi-based receptor. This may be particularly important for danuglipron, as it shows little endosomal signal but significant Golgi and ER signal, suggesting that there are distinct receptor pools.

- Figure 5: The phosphoproteomics is nice although it doesn't provide any physiological insight as it is performed in HEK293 cells. That being said, it is assumed that the observed differences are due to functional selectivity and location bias, and so the statements in the text need to be softened (unless additional experiments are performed to clearly demonstrate that signaling from those different locations is contributing to those profiles). It is surprising how little overlap between the hits on the

upset plot, even looking at the same drug over time. That certainly would argue for location bias playing a role for as the receptor gets internalized, it activates different signaling pathways. This data is broadly supportive of bias between these agonists, but doesn't provide much insight into the underlying mechanisms and little insight into how much of these differences are due to location bias, etc. If a few of these were followed up by Western blot with inhibition of endocytosis or of specific G proteins, etc., that would make it much stronger. Also, please include concentration of drugs used and clarify the sample size (is the $n=3$ for each time point?).

- Figure 6 - I am an unsure as to whether clustering based on Emax and EC50 is the best approach to analyzing the data. While Emax has a more direct relationship to efficacy, EC50 is largely influenced by affinity. So it will naturally lead to higher affinity ligands clustering with one another and vice versa, which honestly isn't as helpful in capturing ligand efficacy. I think some of this problem is seen in Ext Data Fig 10 where this impacts the clustering (all of the EC50s to the left in panel (a)). It would be interesting to see the results of clustering on Emax alone and see if a similar pattern is observed. Ideally affinity could be removed altogether by calculating true efficacies from a Stephenson-based model or taus from the operational model that could be clustered.

- Figure 6c - This same issue with potency affects the cluster center analysis, especially in clusters 1 and 2 where EC50 plays a dominant role. And I'm really just not sure whether this cluster center analysis is the "right" way to analyze the data to determine which signaling pathways could potentially contribute to ADRs.

- Figure 6d - Is the number of ADRs corrected for total number of prescriptions. The association with FDA adverse events is interesting, but more hypothesis-generating than anything else. I hope the authors will follow this up with studies in different pancreatic cells (possibly acinar cells?) to see if those drugs have a specific effect that could contribute to pancreatitis. But, as it stands, it is more of an i

- In the discussion, the authors state "In the present study, we show that location- and pathway-selective signaling profiles impact compound safety." I don't think that is true. They demonstrate a correlation, but it wholly possible that there are other factors that have yet to be assayed (kinetics? Transcription? Something independent of G proteins and beta-arrestins?). It isn't surprising that signaling is a better predictor than structural homology of compounds (which the authors point out later) as changing one residue of a peptide can switch it from an agonist to an antagonist. Overall, my point is that these types of conclusions definitely need to be tempered, stressing the point that an assessment of whole animal physiology is quite far removed from a bunch of assays in HEK293 cells.

- Similarly, the paragraph on page 13 which starts with "In addition to the plasma membrane,..." makes many statements which are not strongly supported by the data. For example, the authors write "For instance, we show that semaglutide and lixisenatide had a lower efficacy for Gi/o than liraglutide and exenatide specifically at the plasma membrane, which was paralleled by a reduction in the incidence of pancreatitis." The tone of this sentence suggests a possible cause and effect relationship which is not supported by the data.

- In the same vein, "Specifically, bARR recruitment at the plasma membrane and retrograde receptor transport, which have been linked to nausea, were revealed to be common features of all peptides, but not the small molecule drug danuglipron. However, the impact of danuglipron-induced GLP-1R activation in the Golgi, but not at endosomes, coupled with extensive rewiring of the cellular phosphoproteome remains unknown." - the data do not support the conclusion as retrograde transport to the Golgi was not clearly demonstrated.

- Given the numerous limitations in the experiments and data interpretation, it would be prudent for the authors to include a section mentioning these limitations and proposing future directions to probe the numerous hypotheses that have been generated from this study. Additionally, suggesting a link between signaling assays of proximal GPCR effectors in HEK293 to complex physiologic events in humans is a correlation. The tone of the discussion should be brought down and the limitations should be mentioned.

- Materials and Methods: Please list all species of constructs used. Please ensure that for all dose-response curves, the time following ligand addition is listed. Is the Extra sum-of-squares F test the appropriate test to determine if the top parameter is statistically distinguishable from the bottom? Many dose-response curves demonstrate insignificant signal in comparison to those obtained for Gas,

GRK, and B-arrestin. The current presentation of the data may impact the clustering analysis.

- Extended Data 4-10: Much of the data shown may mislead readers because the efficacy data is normalized to 100%. For example, in Extended Data 4b, when looking at Gαq activation, there is virtually no activation of Gαq for any of the ligands on the dose-response curve. However, when looking at Extended Data 4c for GLP-1 (7-36), it is denoted as having 100% Gαq activation as these data are normalized – the same data are shown in Extended Data Figure 5a for Gαq. It is more appropriate to normalize the data to maximum signal when an appreciable BRET signal is obtained following ligand activation – the current method of determining activation after comparing the top and bottom parameters from non-linear regression using sum-of-square F-test with a Bonferroni correction may mislead readers.

Minor Comments:

- For the figures, the readability of the concentration-response data is quite poor. It is difficult to place 8 curves on the same small axis, so I understand that limitation, but it would be helpful to have a supplement with them blown up a bit.

- The authors write that “based on these structural differences observed in the residue contact analysis, we selected a combination of drugs...”. I do not believe that this residue contact analysis informed the decision-making of which drugs were chosen as there was little structural insight gained from this experiment, and no information is written describing how these contact analyses informed later experiments. Additionally, the authors use glucagon, GLP-1 (1-37) and liraglutide for further experiments even though no structural data is present for these agonists – this should be clarified.

- The authors write “This endeavor led to the most thorough investigation into the signaling capacity of any GPCR to date, resulting in 480 concentration-response curves comprising 8 drugs, 15 pathways in 4 compartments” – is this true (because high throughput screening campaigns at other GPCRs have produced more data than this)? It would be best to remove the statement that is it the most thorough investigation to date.

- It would be helpful for the authors to provide confocal images demonstrating receptor localization pre and post ligand treatment rather than mGs. It is difficult to make comments about receptor trafficking with the current data.

- For all concentration-response curves in both main and supplemental figures, please include the time point following ligand stimulation for which these data are representing either in the figure or figure legend.

- In Extended Data 5, 7, and 9, it would be helpful to have the coloring of the data when grouping by signaling pathway, be consistent with those depicted in the preceding figures (i.e. each ligand is given a specific color, and that same color is used when showing the data grouped by both ligands and by effector).

REVIEWER COMMENTS

Reviewer #1 (Remarks to the Author):

The ability of GPCRs to signal from different subcellular compartments is now well-established and has been demonstrated for multiple GPCRs. In this manuscript the authors aimed to characterize the signaling properties of GLP-1R at different subcellular locations and in relation to possible adverse side-effects of the drugs. I think the data is compelling and the overall layout is comprehensive and thorough and could be a suitable template for characterization of other receptors in the future. I have several questions and concerns which I am requesting the authors to address in their revised manuscript.

1- Some of the structures used in the analysis in figure 1 are truncated or with engineered stabilizing mutations. Also, some structures were acquired with G proteins coupled with the receptor. Do the authors anticipate this could affect some of the contact maps and downstream analysis or have they taken this into consideration?

As stated in the Materials and Methods section (residue contacts analysis paragraph), for the contact analysis we used refined structure models including a protocol where stabilizing mutations were replaced with WT residues [see <https://docs.gpcrdb.org/structures.html>]. However, all the four active structure complexes in Figure 1 have no such mutations, while the two inactive structures comprise of 14 point mutations for 5VEX (NNC0640) and 13 for 6KJV (PF-06372222). Moreover, all agonist-structures are G_s-bound, minimizing the conformational heterogeneity driven by transducer complexes.

As far as truncations of receptors' segments are concerned, the sequence coverage of the six structures spans from 57% (5VEX, 6KJV) to 84% (6X1A) of the full-length receptor. The differences are mainly driven by the extracellular domain, which accounts for 137 of the missing residues as these are not resolved in the inactive X-ray structures. For this reason, our contact analysis was conducted only on the shared residues among all structures. As detailed in the methods, corresponding residue positions in the structures were identified with the structure-based GPCRdb generic residue numbering system for class B1 (Wootten numbering). This avoids gaps and mismatching after structural alignments of receptors due to missing residues and/or different helix bulges/constrictions among the structures. In total this adds up to 219 shared residues among all 6 structures included in the analysis.

The reviewer is correct that G protein coupling is an important consideration to accurately reflect conformational states and the associated residue contacts. Accordingly, we selected representative structures with the same G protein transducer based on our previous work with identical degree of activation quantified by the intracellular opening spanning the distance between 2x46 to 6x37 (ref. ¹). Briefly, structures in complex with a signaling protein are set as the reference structures for the active state (100%-degree activation). Subsequently, structures with a highly closed conformation are set as the reference structures for the inactive state (0%-degree activation) based on a maximum measured distance between 2x46 to 6x37. The C α atom distance pairs for each structure are compared to the reference structures and the mean distance to the active structures and the mean distance to the inactive structures are calculated. These distances to the reference structure sets are then converted into an "activation score" by subtracting the mean distance to the inactive-state structures from the mean distance to the active-state structures. The activation score is converted into a percentage activation based on the minimum and maximum activation scores for all structures in that class (<https://docs.gpcrdb.org/structures.html?highlight=activation#structure-descriptors>). In our

analysis, all four selected active structures display 100% normalized degree activation. Conversely, the two reference inactive state structures both display 0%-degree activation.

We have added **Supplementary Table 1** related to Figure 1 to better display homo- and heterogeneity among the selected structures and added the aforementioned information to the “Residue contacts analysis” section of the Materials and Methods on page 19.

ID	Ligand	Ligand Type	Ligand Function	PDBid	Method	Resolution	State	Degree active(%)	% sequence coverage	Number of Contact Pairs	Point mutations	Signal protein	Reference	PDB Date
1	NNC0640	SMALL MOLECULE	NAM	5VEX	X-ray	3.0	Inactive	0	57	149	14	nan	10.1038/NATURE22378	17/05/2017
2	PF	SMALL MOLECULE	NAM	6KJV	X-ray	2.8	Inactive	0	57	133	13	nan	10.1107/S2052252519013496	13/11/2019
3	Semaglutide	PEPTIDE	AGONIST	7KIO	cryo-EM	2.5	active	100	83	163	0	Gs(as subunit)	10.1016/J.CELREP.2021.109374	04/08/2021
4	GLP-1	PEPTIDE	AGONIST	6X18	cryo-EM	2.1	active	100	83	171	0	Gs(as subunit)	10.1101/2020.08.16.252585	09/09/2020
5	Exenatide	PEPTIDE	AGONIST	7LLL	cryo-EM	3.7	active	100	79	78	0	Gs(as subunit)	10.1038/S41467-021-27760-0	12/01/2022
6	Danuglipron	SMALL MOLECULE	AGONIST	6X1A	cryo-EM	2.5	active	100	84	140	0	Gs(as subunit)	10.1101/2020.08.16.252585	09/09/2020

2- Figure 1 is a bit hard to read. For example, what is the last column in figure 1a? What is the lower x-axis in the figure 1a? What are the colors in figure 1a represent? Having this information explicitly mentioned in the figure or in the caption would be very helpful.

In response to this comment, we revised the figure, the legend, and simplified our methodological description.

Specifically, we employed a combinatorial optimization algorithm (https://github.com/gecko984/supervenn/blob/master/supervenn/_algorithms.py), which rearranges groupings of columns of contact pairs by the frequency of occurrence (i.e., groupings of shared overlapping contact pairs go first). The column on the right represents the set sizes (N. contact pairs) for each PDB. The numbers on the top (Number of overlapping structures) show how many sets fit into this intersection (i.e., how many PDBs show that particular grouping of contact pairs). The x-axis gives an indication of the proportion of residue contact pairs for each segment across 318 unique contact pairs.

We have revised Figure 1 and the caption accordingly:

“Figure 1. Comparative structure analysis identifies differences in residue contact pairs for GLP-1R agonists. a, Shared contact pairs based on comparative structure analysis (Hauser et al., 2021; Kooistra et al., 2021) represented by a Venn-like diagram (*Supervenn*), where each row is a set of PDB-specific contact pairs, and the overlapping parts (groupings) correspond to shared residue-contact-pairs among a set, sorted by the number of overlapping structures [from six structures (left) to a single structure (right)]. The columns on the right represent the total set sizes (number of residue contact pairs), and the colored ligand structures. The x-axis numbers represent a reference scale as a proportion of the total number of residue contacts shared among all structure combinations (n=318). **b**, Group-specific and unique shared residue contacts mapped onto a representative GLP-1R structure [PDB: 5NX2 (Jazayeri et al., 2017)]. Residues are denoted as circles (Ca) and the non-covalent contacts between residues are denoted as lines. **c**, Polar area diagrams for types of residue contacts (hydrophobic, aromatic, van der Waals, polar and ionic) across ligand-bound structures.” (page 33)

3- Generally, the methods section needs to be updated to include detailed practical information about the measurements. For example, the cell culture media condition, FBS%, amount of DNA used for transfection, duration of expression, concentration of coelenterazine, the buffer that BRET measurement was done in, how is BRET value calculated from the raw data in the plots, if

there are any corrections to the raw data, etc should be included or clearly referenced. With the current level of details, it is hard for other labs to try to replicate the data.

We have added more information in the methods section as requested by the reviewer and included **Supplementary Table 10** for more detailed information regarding the transfection and stimulation conditions.

Regarding the phosphoproteomic analysis:

Overall, this section seemed a bit disconnected from the rest of the manuscript. I suggest the authors incorporate it with the rest of the manuscript and discussion in a more coherent way.

We have improved the flow and description of the phosphoproteomics analysis in the manuscript to better incorporate these data with the rest of the experimental work.

“In light of the location bias that was observed for the small molecule relative to peptide agonists of GLP-1R, we sought to examine the functional consequences of differences in intracellular signal transduction by phosphoproteomics.” (page 9)

4- Does the protein extraction protocol used in the assay recover proteins from internal organelles? Since

For collecting the cells, we used a lysis buffer containing 4% sodium dodecyl sulfate in 50 mM Tris adjusted to pH 8.5 and preheated at 95°C. After the collection, the samples were immediately heated for 10 min at 95°C and sonicated by probe sonication (3 sec on, 3 sec off pulse, 1 min, 30% amplitude). This harsh lysis protocol (adapted from refs. ^{2,3}) allowed for the dissolution of membranes and organelles and thus proteins from internal organelles were also recovered.

These references have now been added to the Phosphoproteomics section of the Methods on page 22.

5- The result that shows enrichment of very different sets of phosphorylated proteins in the presence of different agonists and at different time points is interesting. However, an important control would be performing the experiment with cells not expressing the receptor but treated with the compounds. Have the authors done such a measurement?

In the revised manuscript, we have now included the comparison between untransfected and GLP-1R-expressing cells stimulated with danuglipron and semaglutide at 30 min (**Fig. 5d**). From this control experiment, it is clear that the changes to the phosphoproteome described in our manuscript are receptor-mediated.

“These changes were receptor-mediated and sensitive to an inhibitor of internalization (**Fig. 5d**).” (page 10)

6- From the methods it appears that a phosphatase inhibitor was not used during sample preparation. Usually this is added in such phosphoproteomic analysis to avoid unwanted loss of phosphorylation.

The reviewer is correct that no phosphatase inhibitor was used. The main reason for that choice is the fact that phosphatase inhibitors can interfere with the enrichment of phosphate peptides as discussed in Leutert et al. (ref. ⁴). For collecting the proteomics samples, we lysed the samples in

4% SDS preheated at 95°C and immediately boiled the samples after scraping and inactivation of all enzymes – an approach commonly used for phosphoproteomics sample collection as discussed in Humphrey et al. (ref. ³).

These references are now included in the Phosphoproteomics section of the Methods (page 22).

7- Can the authors comment on how many phosphorylations are on each peptide? In the bioinformatics methods they mention that they allow up to 3 co-occurring phosphorylations for a single peptide in their search parameters? TiO₂ will enrich heavily phosphorylated peptides with greater affinity, so if there is one peptide that maybe occurs in a mono- and di-phospho state did the authors consider that in abundance calculations.

We thank the reviewer for this comment. In our analysis, we detected a median of one phosphorylation event per peptide (average 1.2). As we consider each phosphorylated peptide individually, there should be no bias in the abundance corrections for the mono- and di-phospho states.

8-In the mass spectrometry methods, I'm assuming these parameters are for an MS2 experiment: "30,000 resolution with 27 % NCE, an AGC target was set to 1000%". The NCE is very high and phosphorylations can be ejected during HCD. Have authors verified that this is a safe range of energy for this analysis?

We thank the reviewer for pointing out the instrument-related challenges when performing proteomics analysis. We did not verify the safety of this NCE for this analysis; however, we have used this fragmentation energy for other projects with similar biological material and achieved satisfying results. In addition, the same (or an even higher) fragmentation energy was also chosen for the detection of phosphopeptides by other research groups, as seen in Martinez-Val et al. (ref. ⁵), Salovska et al. (ref. ⁶) and Kitata et al. (ref. ⁷). Thus, the NCE chosen here in this study is also within the range of NCEs used for phosphoproteomics analyses and is in agreement with what is currently used in the field.

8- When the authors did comparative quantification between samples, are they comparing the abundance of a specific peptide, modification state, and charge state? The charge and modification state will change the ionization efficiency a great deal and this will impact the calculated abundance calculation. For example, are they comparing some peptide like N-GER(pT)SIP-C in the 2+ charge state directly between different conditions?

For our analysis, we only compare phosphopeptides with the same positional phosphosites. Different charge states (precursors) were not treated as different peptides.

9- Also, is the abundance of a peptide defined as number of scans? Or the actual area under the curve of ion current for a specific peptide on the LC timescale?

The abundance of a peptide is defined as the actual area under the curve.

This has been added to the methods:

"Peptide quantification was performed based on the fragment elution profiles from the MS2 level (area under the curve)." (page 23)

10- The written mass spectrometry methods are pretty well-written. I assume the Lumos was set

into peptide mode and the trapping pressure was "low" (2 mTorr?). Please specify that parameter as well. Please also specify the source used on the front end of the mass spec, e.g. is it a HESI, NanoFlex, etc.?

We have extended the materials and methods section for the mass spectrometry methods with the following paragraph:

"For data acquisition, the instrument was set to peptide mode, with a trapping pressure set as low (2mTorr). An EASY-Spray Source (Thermo Fisher Scientific) was used and the column temperature was maintained with the integrated, temperature-controlled heater at 55°C throughout the experiment. For the second experiment, the column was heated at 50°C following the manufacturer's instructions (IonOpticks)." (page 22)

Reviewer #2 (Remarks to the Author):

Authors

GPCRs classically signal from the plasma membrane, but several studies indicate that they can maintain signaling from intracellular compartments, largely in the endosomal pathway. So that authors ask whether membrane permeable small molecule agonists of the GLP1 receptor have different cellular signaling effects than do peptide agonists, which do not enter cells. Not surprisingly the answer is 'yes' as is established by the authors in studies carried out using an array of techniques in HEK293 cells. In addition, the small molecule agent under study (danuglipron) was shown to be less biased toward beta-arrestin mediated signaling than peptide agents. The authors extrapolate from their HEK293 results to some conclusions regarding likelihood of adverse drug reactions of the agents. There are a number of issues that need author attention:

--- The gist of this paper is that (from the authors' Discussion) "Stimulation of GLP-1R with peptide agonists and cell permeable small molecule agonists results in different subcellular activation." This is perhaps not surprising. I think that it is problematic to extrapolate, however, from studies in HEK293 cells with overexpressed signaling proteins to predictions about responses and adverse drug reactions in human patients. So, although the data presented are of interest, the authors need to guard against overinterpretation.

The reviewer highlights one important aspect of our paper relating to the difference between peptide and small molecule activity. However, the gist of the paper is too simplified as described. Specifically, we describe for the first time that we can now measure compartmentalized signaling for several pathways and organelles which may help to unravel the intricate details of GPCR signaling with unprecedented resolution. The reviewer is correct that we need to guard against overinterpretation, and we have made changes throughout the manuscript to tone down any phrasing that suggests we are proposing causal relationships between the differences observed in spatiotemporal signaling activity and the reported adverse drug reactions. We foresee this paper as being a steppingstone for larger studies that will investigate whether the correlations we identified between signaling neighborhoods and ADRs can provide useful guidance for drug discovery and development in the future.

We have specifically addressed this by adding the following sentence to the discussion:

"Importantly, the firm establishment of causal relationships between transducer signatures and ADR risk requires future evaluations on a larger scale." (page 13)

--- We are entering into an enhanced data sharing era, which requires comprehensive data sharing of all data types. In proteomics, data sharing has been the norm for more than 10 years. In general, detailed sequence data is shared for all protein mass spectrometry data, usually via one of the members of the ProteomeXchange Consortium. In Europe, this would be PRIDE at <https://www.ebi.ac.uk/pride/>. Upon deposition, PRIDE provides a link and reviewer login credentials to allow review of the data. The authors have indicated they they have so deposited the data, but inserting the given number gave a message, "Sorry, no projects found for search term PXD037472 and the current set of active filters." More significantly, the data have not been provided in curated form, e.g. in a spreadsheet, reporting the abundance every phosphopeptide (or every phospho-site) in every sample, along with calculations used to create Figure 5. The phosphoproteomic data sound interesting, but without access to the data, it is impossible to assess data quality or to replicate the basic calculations.

We sincerely apologize to the reviewer. The data has been deposited on PRIDE as stated in the paper, however, the reviewer login information was forgotten, which is as followed:

Username: reviewer_pxd037472@ebi.ac.uk
Password: J4YnJVyr

We totally agree with the reviewer that data sharing in proteomic research is important and the data should be as easy to access as possible. With that in mind, we have also uploaded the newly generated proteomics dataset to the same PRIDE accession number, making it even easier for anybody who wants to access any of the two datasets collected as part of this project. We can ensure the reviewer that the data will be made publicly available upon acceptance of the paper. In addition, we have also created several spreadsheets containing quantitative information from all phosphopeptides identified in our experiment (**Supplementary Tables 3-7**). This also includes the new experiment performed during the revision of this article.

---Focusing on the phosphoproteomics some information was missing from the Methods section. The phospho-proteomic studies seem to confirm the conclusion that the small molecule agent used produces different signaling responses than one of the peptide agonists. Nevertheless, the presentation of phosphoproteomic data was superficial and did not even tell us any of the phosphorylation sites that were changed in response to the agents.

We have added the individual phosphorylation sites to the subpanel in question which is now found in **Supplementary Fig. 8**. In addition, we also provide a spreadsheet with all quantitative data including the phosphorylation sites (**Supplementary Tables 4-7**).

--- The proteomics methods state "The experiment was performed in biological triplicates." That means that there were 45 total samples... 5 time points for each of three treatments in triplicate. 5 x 3 x 3. Please confirm. Having a spreadsheet with all phosphopeptides in all samples would have clarified this.

We thank the reviewer for this comment. A spreadsheet related to these samples was added to **Supplementary Table 3** (referred to on page 9). In total, 39 phosphoproteomics samples were analyzed in the first analysis and 27 in the second.

The first analysis (Fig. 5b,c,e and Supplementary Fig. 8) consisted of 4 time points (5, 15, 30, 60 min) and 3 treatments (vehicle, semaglutide and danuglipron) performed in triplicate in addition

to vehicle treatment at time point 0 also performed in triplicate. For the second analysis (Fig. 5d), a total of 27 samples consisting of untransfected cells, GLP-1R-expressing cells or GLP-1R-expressing cells treated with internalization inhibitor were treated with vehicle, semaglutide or danuligpron for 30 min in triplicate.

--- Authors should be more explicit about phosphopeptide quantification. Apparently they used a label-free method, but the principle of the quantification is vague. Presumably, the method calculated the area under the MS1 curve to get integrated intensity values. Please provide this information. For quality control, it would be helpful to show concordance between two peptides with the same phospho-site (e.g. from different charge states or missed cleavages). Availability of a spreadsheet with the intensities for each peptide in each sample would have been helpful. It was not among the Supplementary Files.

The reviewer is correct that a label-free method was used for quantification. However, quantification was performed on the MS2 level as this is standard for DIA-based quantification. We have amended the materials and methods section to clarify this. We have also created a spreadsheet containing the normalized intensities from all phosphopeptides (**Supplementary Tables 5 and 7**).

--- It is a disappointing that a labeling method was not employed such as TMT. Since this technique directly compares peptide abundances in all samples, the data would have been stronger and more phosphopeptides could have been reported. Matching peptide quantification from different MS runs is always problematic.

Due to the large number of individual samples for this phosphoproteomics study ($n=39$), we were unable to fit them all in a single TMT plex, as the largest commercially available one is 18-plex at the moment. This would have required splitting the samples into different TMT plexes, which could reduce the data completeness, due to the stochastic nature of DDA sampling, required for TMT data acquisition.

--- In the phosphoproteomic data, it would be helpful to know if any of the agonists increased phosphorylation of the COOH-tail of the GLP1 receptor, particularly at sites thought to be targeted by GRKs.

We thank the reviewer for this comment. In our analysis, we carried out a search for the COOH-tail of GLP-1R, but we were not able to quantify this specific peptide. This observation is in line with data from the Peptide Atlas, indicating that detection of the COOH-tail of GLP-1R (or any peptide from GLP-1R) is not trivial.

--- Figure 5d should give the specific phosphorylation site and the phosphopeptide sequences listed in the legend for transparency.

The specific phosphorylation sites have been added to this figure which has now been moved to **Supplementary Fig. 8** to make room for the new phosphoproteomics analysis requested by reviewers 1 and 3.

--- Figure 5e is obscure. What would be more helpful would be a report of the sequence motifs that would be formed from differentially phosphorylated sites. These motifs can then be mapped to kinases using data from a recent paper published in Nature (Nature 613, 759–766 (2023). DOI: 10.1038/s41586-022-05575-3). There are a number of programs that can generate the associated

kinase target sequence logos such as PTM-Logo (<https://hpcwebapps.cit.nih.gov/PTMLogo/>). The black box nature of the KSEA analysis undermines confidence in the conclusions.

We thank the reviewer for directing us to this publication. As this paper was published after the submission of our manuscript, it was not possible for us to consider it in our analysis at the time. However, we fully acknowledge the value of such a novel analysis and the systematic evaluation of kinase motifs. Accordingly, we append to our manuscript a novel kinase substrate enrichment analysis in **Fig. 5e**.

--- The authors pick out the Ephrin type-B receptor 4 and the AMP-dependent kinase as two protein kinases that are involved in ER stress. Regulation of the ER stress/UFR mechanism is not the primary role of these kinases however and the implication in the paper that Danuglipron impairs ER function is not well supported.

Following the reviewer's suggestion to use the atlas for Ser/Thr kinase motifs, we have removed this passage and focused on the kinetic differences that support differences in subcellular signaling.

--- The plots shown in Figure 5b are a bit problematic. According to the diagram in Figure 5a, the same vehicle sample was used for calculation of both the y axis value ($\log_2(\text{Semaglutide}/\text{Vehicle})$) and the x-axis value $\log_2(\text{Danuglipron}/\text{Vehicle})$. As a consequence, the nice correlations shown can be called into question since the same term (vehicle) is used in both x and y axis. However, the main conclusion that the responses differ between the two agents is not in question, so this is not a crucial point, although worth clarifying in the paper. In addition, it is not clear from the presentation that independent time control vehicle samples were collected for each time point. Data sharing would have helped sort this out.

We thank the reviewer for this observation. Yes, the same control samples (time-matched) were used for both x- and y-axes. However, this does not affect the correlation as we just introduce a constant factor to both axes, hence we only perform a linear transformation. We have now clarified this in the legend for **Fig. 5b**. We also provide an Excel table containing the source data for this plot (**Source data**).

--- It seems questionable whether studies with overexpressed GLP1-R and effectors in HEK293 cells successfully mimic physiological responses with signaling proteins at endogenous levels. The type of study described in the paper can tell us if a given protein-protein interaction CAN occur and whether a drug CAN effect a given response, but does not establish whether these responses DO occur in native tissue. Authors need to address this limitation and consider animal studies that may strengthen the conclusions. (This is a problem with the whole promiscuous GPCR-G protein coupling story.)

We fully acknowledge that the data presented here demonstrate how specific drug-receptor complexes *can* result in various signaling events and does not mean these interactions will necessarily occur in native tissues. We have addressed this limitation in the discussion.

Further investigation *ex vivo* with endogenous protein expression in a physiologically relevant tissue model could help address this question, but interindividual differences or changes in protein stoichiometry could influence the outcome of these experiments. This limitation supports the use of assays where complex stoichiometry has been optimized like the one described herein for probing the GPCR transducerome. Several studies have demonstrated that while promiscuous GPCR-G protein coupling does occur for many receptors, it does not happen for all. Our findings

are in line with the ability of GLP-1R to engage several G protein subtypes, but there are many examples of GPCRs that have a higher degree of selectivity or that do not engage G proteins at all (refs. ⁸⁻¹²).

--- The abstract is too generic. It sounds like it was written with ChatGPT or similar. "...translating signaling neighborhoods into functionally distinct cellular outcomes and clinical consequences." is not going to be understood by many. It seems like the abstract should say that cell permeant small molecule agonists have different cellular effects than do peptidic agents that cannot enter the cell. This is more clearly stated in the Discussion in the section, "Stimulation of GLP-1R with peptide and small molecule agonists results in selective subcellular activation."

Our finding that cell permeant small molecule agonists have different cellular effects than biologics is indeed an important observation of our manuscript, but an oversimplification. Our analysis also allowed us to associate less obvious differences among peptide agonists of GLP-1R with ADRs. We have modified the passage in the abstract highlighted by the reviewer to make this clearer:

"These findings, together with comparative structure analysis, time-lapse microscopy and phosphoproteomics, reveal unique signaling signatures for GLP-1R agonists at the level of receptor conformation, functional selectivity and location bias, thus associating *signaling neighborhoods* with functionally distinct cellular outcomes and clinical consequences." (page 3)

--- The statement, "To our surprise, we also observed a convergence and accumulation of mGs positive vesicles to the Golgi apparatus suggesting that GLP-1R may engage in endosome-to-Golgi retrograde transport..." should be reconsidered. Every plasma membrane integral membrane protein is translated on the rough ER and must advance through the Golgi and through secretory vesicles to get to the plasma membrane. So the presence of signal emanating from any of those compartments does not imply retrograde transport.

There seems to be a misunderstanding. We were not referring to protein export to the plasma membrane, which would be, as the reviewer correctly states, not surprising. Rather, we demonstrate, by BRET and confocal microscopy, that an agonist-bound receptor translocates in its active form from the plasma membrane to the Golgi. Specifically, we demonstrate that labelled receptor traffics from the plasma membrane to endosomes and the Golgi apparatus in **Supplementary Figs. 5 and 6** and that this coincides with data obtained using the biosensor mGs that measures the active conformation of the receptor capable of activating G_s (**Figs. 3, 4, Supplementary Fig. 4 and Movie 1**).

--- Extrapolation from results with overexpressed GLP1-R and effectors in HEK293 cells to likelihood of adverse effects in human patients seems risky. Animal studies would seem more useful. It is advised to remove the whole ADR element from this paper or to mention it only briefly. Its inclusion reduces confidence in the main conclusions of this paper.

We have toned down the wording in the manuscript that incorrectly suggests a causal relationship between signaling neighborhoods and ADRs, but the correlation between the two suggests that this is an important aspect of drug development that requires further investigation. Our analysis of data from the FDA shows a higher likelihood of certain ADRs are overrepresented in liraglutide- and exenatide-treated patients. Signaling signatures might be a relevant complement to further improve predictions in the future.

In response to the reviewer, we have modified the title of the manuscript, any words in the manuscript that imply causation and added a paragraph to the discussion relating to limitations and future perspectives (page 13).

--- The disclosure is incomplete. It needs to state whether Domain Therapeutics is working on agents that target GLP1-R. This is unclear from the DT web site.

Although Domain Therapeutics does not disclose its targets, it is focusing its activity on GPCRs involved in immuno-oncology and GLP-1R does not fall under this description.

--- Minor: There seems to be a lot of pseudo-systems-biology jargon like the term 'rewiring' which is used in so many ways that it is not clear what is meant in this paper.

Rewiring was used to describe the difference between incretin mimetic-induced changes to the phosphoproteome versus small molecule treatment. Based on the comment by the reviewer, we have chosen to replace rewiring with changes to the phosphoproteome. That being said, this study has employed systems biology methods, such as hierarchical clustering, *k*-means cluster centers analysis and phosphoproteomics, to more comprehensively characterize the differences between GLP-1R agonists and the choice of wording reflects the complexity of this approach.

Reviewer #3 (Remarks to the Author):

This is a very well-performed manuscript that characterizes drug efficacy and location bias across a range of agonists at the GLP-1R, a very important drug target in diabetes and obesity (probably the most important drug target). The experiments are rigorously performed by groups that have developed the enhanced bystander BRET (ebBRET) platform that allows an assessment of pathway activation in a location-specific manner. The experiments are well-performed and the number of assays performed very impressive (especially looking across 15 assays at 4 locations and 8 conditions). The overall conclusion (that these drugs have wildly different patterns of signaling, especially based on location) is supported by the data. But the weakness of the manuscript are the conclusions/claims that are not fully supported by the data (such as the adverse drug reactions) and data that really is not central to the major claims and is sort of supportive but not fleshed out well (phosphoproteomics). So I think that the manuscript largely requires a toning down of claims and some different types of analyses.

Major Comments:

- Figure 1 – The comparative structure analysis is nice, but doesn't add much unless those insights are used to guide experiments. I think this may be better served as a supplement as no real conclusions arise from it other than that they have different binding modes (not surprising).

We believe the structural analysis is an important puzzle piece to link receptor-structure-function to transducer-specific engagements in different cell compartments (i.e., the observation that ligand-specific receptor residue engagements have transducer and compartment explicit consequences). This is an important observation, which might guide future drug discovery activities both for GLP-1R and for GPCRs in general. It is also intriguing (and not obvious) that peptides sharing higher homology like semaglutide and GLP-1 (7-36) have such different contact networks and sets the stage for our investigation into their signaling neighborhoods.

While we could not conclude on any direct link from the structural changes to transducer coupling in different compartments, we can conclude that the binding of these different ligands promotes

both distinct conformational rearrangements and differential transducer signaling. With an ever-increasing number of GLP-1R structures being published, we may eventually be able to design experiments that address which specific contact networks drive differential signaling. Our study sets the stage for such future analyses.

- Figure 2 – This is an issue with Figure 6 as well, but I don't think the comparisons across EC50 in (d) are not insightful and, if anything, misleading. Those similarities are primarily related to affinity and not efficacy, and since affinity varies considerably between different peptides and small molecule agonists, the obtained results do not provide meaningful insights. While the Jaccard index provides some quantitative assessment of similarity, an assessment of bias would be better performed using the approaches recently outlined in IUPHAR community guidelines (<https://pubmed.ncbi.nlm.nih.gov/35106752/>).

The point of the reviewer is well taken regarding potency and affinity. Focusing on the drugs we included in our study, potency and efficacy differ across transducers and across compartments. This is exemplified in the new **Supplementary Figure 18**, where we measure the equality for efficacy and potency across transducers and compartments using the Gini coefficient. As highlighted by the reviewer, major differences can be seen in efficacy. However, important differences were still observed when looking at potency suggesting that affinity is not the only driver for agonist potency. This is an important reason for including potency in our clustering analysis.

Bias calculations are only possible with a reference ligand and pathway severely limiting the number of meaningful comparisons that can be made. Moreover, because potencies can vary by several orders of magnitude, a low efficacy agonist that is very potent can appear to be biased relative to a higher efficacy agonist with a lower potency. This is further complicated by the fact that having a reference ligand across all pathways and in all compartments is not possible for any GPCR currently – the analysis we present in this paper has thus far only been carried out at GLP-1R. In an attempt to bring potencies and efficacies on the same scale, we calculated the z-scores across drug treatments in order to feed the clustering algorithm with as much data as possible.

In line with the IUPHAR community guidelines, we have chosen not to include bias calculations given the lack of balanced reference ligand in every tested pathway and compartment (see 3.1.1 Problems and pitfalls in the IUPHAR review 32). To exemplify the caveats of this to the reviewer, we have included two examples of where bias calculations do not meet the mark (see below). The first example demonstrates how lixisenatide (high potency, low efficacy) appears biased towards G_z vs. G_s relative to the reference ligand GLP-1 (7-36), while exenatide (high efficacy, low potency) appears to be biased away from G_z . This striking difference is driven by potency and masks important differences in the pharmacology of these ligands. Probably the most obvious difference which would never come across in a bias analysis is the observation that danuglipron has an effect in the ER on G_s signaling, but none of the other ligands do. Since the reference ligand [GLP-1 (7-36)] is not active for G_s signaling in the ER, it would not be possible to calculate bias factors for ligands like Danuglipron that do have activity, thereby preventing the comparison of pathways in different compartments.

- Figure 3 – From the text and the figure it is not clear to me whether the Gs is only noted at the Golgi. Please show merged images to better demonstrate colocalization of the location markers and mini Gs constructs.

We have included merged images as insets in **Figure 3** to show the colocalization of mGs with the different compartment markers. It should be noted that the specific marker that we have chosen for the Golgi, Giantin, is targeted to specific subdomains of the Golgi apparatus (i.e., cis-medial rims) resulting in an incomplete colocalization that is nevertheless sufficient to sensitively detect colocalization when used in enhanced bystander BRET assays.

- Figure 4 – The time course of the signal in the Golgi would be interesting to note. Is there signal in the Golgi before 30 minutes? It is important to differentiate the retromer based process of endosome to Golgi as opposed to what has been seen at the beta 1 adrenergic and adenosine A1 receptors where there are distinct pools of Golgi-based receptor. This may be particularly important for danuglipron, as it shows little endosomal signal but significant Golgi and ER signal, suggesting that there are distinct receptor pools.

The reviewer brings up an interesting point that we have now addressed in our revision with confocal microscopy and BRET analysis. Using a cell impermeable SNAP substrate to label SNAP-GLP-1R at the cell surface, we observe receptor trafficking from the plasma membrane to endosomes and to the Golgi (**Supplementary Fig. 5**). We also quantified the kinetics of our previous dataset with mRuby2-mGs and observe a steady accumulation of mGs in the Golgi following semaglutide stimulation (**Supplementary Fig. 4**). In parallel, we analyzed the netBRET of luciferase-tagged GLP-1R in combination with different compartment markers. While the BRET is not directly comparable in these different experimental paradigms, we can conclude that there is also Golgi-localized GLP-1R in addition to the plasma membrane under basal conditions suggesting that there are distinct receptor pools and different modes of receptor regulation for peptide and small molecule agonists (**Supplementary Fig. 6**).

- Figure 5: The phosphoproteomics is nice although it doesn't provide any physiological insight as it is performed in HEK293 cells. That being said, it is assumed that the observed differences are due to functional selectivity and location bias, and so the statements in the text need to be softened (unless additional experiments are performed to clearly demonstrate that signaling from those different locations is contributing to those profiles). It is surprising how little overlap between the hits on the upset plot, even looking at the same drug over time. That certainly would argue for location bias playing a role for as the receptor gets internalized, it activates different signaling pathways. This data is broadly supportive of bias between these agonists, but doesn't provide much insight into the underlying mechanisms and little insight into how much of these differences are due to location bias, etc. If a few of these were followed up by Western blot with inhibition of endocytosis or of specific G proteins, etc., that would make it much stronger. Also, please include concentration of drugs used and clarify the sample size (is the n=3 for each time point?).

Following the suggestion by the reviewer, we conducted additional phosphoproteomic analyses with and without the internalization inhibitor (Dyngo-4A) following 30 min of stimulation with either danuglipron or semaglutide (**Fig. 5d**). Both semaglutide and danuglipron have been previously shown to induce the internalization of GLP-1R (ref. ¹³) and preincubation with the internalization inhibitor affected the magnitude of agonist-induced phosphorylation events implicating location bias in GLP-1R signaling. We should emphasize that in contrast to our BRET experiments, phosphoproteomic analysis was carried out in the absence of transducer overexpression. As highlighted by the reviewer, little overlap was observed between semaglutide and danuglipron

when looking at significant phosphopeptides. In fact, danuglipron induced more phosphorylation events than semaglutide in both rounds of phosphoproteomics. This is interesting, given the greater degree of kinase activity linked to semaglutide administration. In the new kinase substrate analysis requested by reviewer 2, we observed trends in kinase activity for semaglutide that were more consistent with receptor-mediated endocytosis than danuglipron. Taken together, the new phosphoproteomic data and kinase analysis suggest that danuglipron is capable of inducing phosphorylation events that are endocytosis-mediated or driven by passive/active transport.

We have discussed the kinetic differences in kinase activity in the following passage:

“Importantly, kinase kinetics differed between small molecule and peptide treatments. Semaglutide resulted in trends that aligned with sustained signaling following receptor-mediated endocytosis with maximal changes observed after 30-60 min. In contrast, danuglipron-treated showed distinctly different kinetics, demonstrating that different compounds elicit distinct compartment-specific response signatures.” (page 10)

The concentration of drugs used (1 μ M) is found in the Methods section on page 21.

- Figure 6 - I am unsure as to whether clustering based on Emax and EC50 is the best approach to analyzing the data. While Emax has a more direct relationship to efficacy, EC50 is largely influenced by affinity. So it will naturally lead to higher affinity ligands clustering with one another and vice versa, which honestly isn't as helpful in capturing ligand efficacy. I think some of this problem is seen in Ext Data Fig 10 where this impacts the clustering (all of the EC50s to the left in panel (a)). It would be interesting to see the results of clustering on Emax alone and see if a similar pattern is observed. Ideally affinity could be removed altogether by calculating true efficacies from a Stephenson-based model or taus from the operational model that could be clustered.

We have performed the clustering based on efficacy alone and by and large, it resembles the clustering that we obtain when we also include potency (**Supplementary Fig. 19c**). We wish to direct the reviewer to our previous response regarding potency, efficacy and calculations that attempt to integrate the two into a single meaningful value. In short, we hope to have convinced the reviewer that while potency is influenced by affinity, the two are clearly not equivalent and vary across pathways and compartments. In addition, the mere fact that potencies can vary over several orders of magnitude argues for not using the operational model.

- Figure 6c – This same issue with potency affects the cluster center analysis, especially in clusters 1 and 2 where EC50 plays a dominant role. And I'm really just not sure whether this cluster center analysis is the “right” way to analyze the data to determine which signaling pathways could potentially contribute to ADRs.

While this approach has never been used in the context of GPCR signaling, cluster center analysis is a statistically robust approach to identify and associate patterns in one analysis with a pattern based on clinical data (refs. ¹⁴⁻¹⁶). Here, we find that GPCR transducer patterns and ADRs are partly aligned; incentivising further research. However, the current data does not allow for us to imply causation and we have carefully reworded all statements to this effect.

In particular, we have emphasized this important limitation in the discussion with the following sentence:

“Importantly, the firm establishment of causal relationships between transducer signatures and ADR risk requires future evaluations on a larger scale.” (page 13)

- Figure 6d – Is the number of ADRs corrected for total number of prescriptions. The association with FDA adverse events is interesting, but more hypothesis-generating than anything else.

Adverse reactions (ADRs) have been estimated from reports submitted to the FDA Adverse Event Reporting database (FAERS) by healthcare professionals by transformation into a log likelihood ratio (LLR) (refs. ^{17,18}) and which corrects for the prevalence of a given drug and the frequency of an event across drugs.

As described, the adverse reactions reports were filtered as in OpenTargets (<https://platform-docs.opentargets.org/drug/pharmacovigilance>) (ref. ¹⁸) as follows:

- Only reports submitted by health professionals
- Exclude reports that resulted in death
- Only drugs that were considered by the reporter to be the cause of the event
- Remove blacklisted events curated manually to exclude uninformative events.

The significant drug-ADR pairs are then evaluated using the Likelihood Ratio Test (LRT) as previously described (ref. ¹⁷). The significance of a given drug-ADR is implicitly corrected by how often a drug is found in a report and how often an event is reported across drugs. This is in order to prevent drug-ADR associations from being biased by overrepresented ADRs (e.g., headache, nausea) or drugs (e.g., paracetamol, ibuprofen). In order to assess significance, an LRT critical value for every drug is calculated using an empirical Monte Carlo simulation (ref ¹⁸).

We do, however, agree with the reviewer that this analysis is hypothesis-generating and sets the stage for additional studies exploring the possible causalities between signaling profiles and adverse effects.

-I hope the authors will follow this up with studies in different pancreatic cells (possibly acinar cells?) to see if those drugs have a specific effect that could contribute to pancreatitis. But, as it stands, it is more of an i

We thank the reviewer for the comment and agree that transferring these tools to relevant cell types is an important and necessary next step.

- In the discussion, the authors state “In the present study, we show that location- and pathway-selective signaling profiles impact compound safety.” I don’t think that is true. They demonstrate a correlation, but it wholly possible that there are other factors that have yet to be assayed (kinetics? Transcription? Something independent of G proteins and beta-arrestins?). It isn’t surprising that signaling is a better predictor than structural homology of compounds (which the authors point out later) as changing one residue of a peptide can switch it from an agonist to an antagonist. Overall, my point is that these types of conclusions definitely need to be tempered, stressing the point that an assessment of whole animal physiology is quite far removed from a bunch of assays in HEK293 cells.

We have tempered the conclusions as suggested by the referee. For the passage in question, we have replaced “impact” with “correlate with.”

- Similarly, the paragraph on page 13 which starts with “In addition to the plasma membrane,…” makes many statements which are not strongly supported by the data. For example, the authors

write “For instance, we show that semaglutide and lixisenatide had a lower efficacy for $G_{i/o}$ than liraglutide and exenatide specifically at the plasma membrane, which was paralleled by a reduction in the incidence of pancreatitis.” The tone of this sentence suggests a possible cause and effect relationship which is not supported by the data.

We agree with the reviewer regarding this statement and have reworded it as follows:

“For instance, we show that semaglutide and lixisenatide had a lower efficacy for $G_{i/o}$ than liraglutide and exenatide specifically at the plasma membrane and coincidentally have a lower incidence of pancreatitis.”

- In the same vein, “Specifically, β ARR recruitment at the plasma membrane and retrograde receptor transport, which have been linked to nausea, were revealed to be common features of all peptides, but not the small molecule drug danuglipron. However, the impact of danuglipron-induced GLP-1R activation in the Golgi, but not at endosomes, coupled with extensive rewiring of the cellular phosphoproteome remains unknown.” – the data do not support the conclusion as retrograde transport to the Golgi was not clearly demonstrated.

We have demonstrated that retrograde transport of GLP-1R can take place following stimulation with semaglutide (**Fig. 3** and the new **Supplementary Fig. 6**). The reviewer was correct in highlighting this sentence because it is in fact β ARR recruitment and *receptor endocytosis* which have been linked to nausea and while we have observed the progression into retrograde transport for the high affinity agonist semaglutide, we have no evidence that other peptides behave this way or that retrograde transport itself coincides with an increased incidence of nausea. We have reworded the passage in question as follows:

“Specifically, β ARR recruitment at the plasma membrane and receptor endocytosis, which have been linked to nausea, were revealed to be common defining features of all incretin mimetics, but not the small molecule drug danuglipron.”

The endocytosis data are inferred from Fig. 2 which shows BRET between Rluc8-mGs and rGFP-FYVE. mGs should not accumulate at endosomes unless the receptor is present since it is a sensor for the active receptor conformation that can engage and activate Gs.

- Given the numerous limitations in the experiments and data interpretation, it would be prudent for the authors to include a section mentioning these limitations and proposing future directions to probe the numerous hypotheses that have been generated from this study. Additionally, suggesting a link between signaling assays of proximal GPCR effectors in HEK293 to complex physiologic events in humans is a correlation. The tone of the discussion should be brought down and the limitations should be mentioned.

We have now included a part in the discussion relating to limitations and future directions for our study:

“Moving forward, it would be important to extend the concept of signaling neighborhoods to primary human tissues to evaluate whether the profiles are conserved. Also, secretin receptor family crosstalk or interpatient variability would be interesting parameters to consider to fully comprehend the physiological impact of pathway-selective compartmentalized signaling. Another intriguing aspect of GLP-1R signaling that deserves attention is the potential for kinetic bias of drugs acting on this receptor as this may modulate efficacy and potency^{45, 46}. Future research should therefore explore the consequences of different dosing regimens and pharmacokinetics of

GLP-1R agonists in vivo on both signaling profiles and biological outcomes. Importantly, the firm establishment of causal relationships between transducer signatures and ADR risk requires future evaluations on a larger scale.” (page 13)

- Materials and Methods: Please list all species of constructs used. Please ensure that for all dose-response curves, the time following ligand addition is listed. Is the Extra sum-of-squares F test the appropriate test to determine if the top parameter is statistically distinguishable from the bottom? Many dose-response curves demonstrate insignificant signal in comparison to those obtained for Gas, GRK, and B-arrestin. The current presentation of the data may impact the clustering analysis.

All constructs used in this manuscript were based on the human sequences. This has been added to the materials and methods (page 16).

Choosing the most appropriate way to analyze our dataset was not an easy task, but we chose to apply the extra sum-of-squares F test to compare the top and bottom plateaus of all non-linear fits – a difference being a prerequisite for efficacy and efficacy itself being a prerequisite for potency. In addition to this, we applied stringent criteria to reduce the number of false positives including multiple testing correction and curation of potencies that were estimated to be outside the range of concentrations tested. Please see our response below with regard to the presentation of the data.

- Extended Data 4-10: Much of the data shown may mislead readers because the efficacy data is normalized to 100%. For example, in Extended Data 4b, when looking at Gαq activation, there is virtually no activation of Gαq for any of the ligands on the dose-response curve. However, when looking at Extended Data 4c for GLP-1 (7-36), it is denoted as having 100% Gαq activation as these data are normalized – the same data are shown in Extended Data Figure 5a for Gαq. It is more appropriate to normalize the data to maximum signal when an appreciable BRET signal is obtained following ligand activation – the current method of determining activation after comparing the top and bottom parameters from non-linear regression using sum-of-square F-test with a Bonferroni correction may mislead readers.

As the reviewer is well aware, normalizing drug responses to a reference ligand is common practice in receptor pharmacology. However, this is usually limited to well-defined receptor readouts and prevents a complete investigation into the true signaling capacity of a receptor of interest especially for different types of drugs (biologics vs. small molecules). In our recent publication in eLife, we also normalized signaling responses to reference receptors for 100 GPCRs (ref. ⁸). While this can circumvent certain issues when interpreting efficacy, it may underestimate the importance of smaller responses and cannot be used in compartments where pathway engagement has not been benchmarked for any receptor. Here, we could not choose reference receptors for the 15 signaling pathways originating from 4 compartments because this analysis has never been done before. Complicating matters further is the fact that we cannot compare the absolute BRET signal between pathways (i.e., the strength in signal between p63RhoGEF and Rap1Gap). When we compare several ligands to each other across different pathways, differences become more meaningful. For this reason, the efficacies plotted in Figure 2 and Supplementary Figures 9, 12 and 15 may be seen as a more quantitative rank order of all tested ligands. Consequently, we do not interpret the importance of specific pathways in specific compartments, but rather the collective impact of drug signaling signatures and their apparent correlation with adverse drug reactions.

To improve the presentation of our data, we have opted to only include the normalized fits (min-max) in **Figure 2** and **Supplementary Figures 9, 12 and 15** and to include separate supplementary figures with every raw concentration response plotted individually for all drugs, pathways and compartments (**Supplementary Figures 2, 10, 13 and 16**).

Minor Comments:

- For the figures, the readability of the concentration-response data is quite poor. It is difficult to place 8 curves on the same small axis, so I understand that limitation, but it would be helpful to have a supplement with them blown up a bit.

We have now included miniaturized plots for every drug across all pathways and compartments in **Supplementary Figures 2, 10, 13 and 16**.

- The authors write that “based on these structural differences observed in the residue contact analysis, we selected a combination of drugs...”. I do not believe that this residue contact analysis informed the decision-making of which drugs were chosen as there was little structural insight gained from this experiment, and no information is written describing how these contact analyses informed later experiments. Additionally, the authors use glucagon, GLP-1 (1-37) and liraglutide for further experiments even though no structural data is present for these agonists – this should be clarified.

The reviewer is correct in that some of the drugs used for further investigation have not been solved in complex with GLP-1R and therefore could not be included in the structural analysis. We included additional ligands with varying degrees of homology to the peptide ligands from the structural analysis including GLP-1 mimics like liraglutide and the precursor peptide glucagon as well as variants of exenatide from the Gila monster like lixisenatide. We have reworded the sentence to increase its clarity and link with the subsequent experiments:

“Based on the structural differences observed in the residue contact analysis, we selected a series of homologous peptide agonists (GLP-1 and exendin-4 analogs) in addition to the small molecule agonist (Supplementary Table 2) to investigate the pathway selectivity of GLP-1R at the plasma membrane using the effector membrane translocation assay (EMTA) based on enhanced bystander BRET (ebBRET) in living cells.” (pages 6-7)

- The authors write “This endeavor led to the most thorough investigation into the signaling capacity of any GPCR to date, resulting in 480 concentration-response curves comprising 8 drugs, 15 pathways in 4 compartments” – is this true (because high throughput screening campaigns at other GPCRs have produced more data than this)? It would be best to remove the statement that is it the most thorough investigation to date.

The reviewer is correct that HTS campaigns have included more ligands for a single or few receptors/pathways. However, screening campaigns at other GPCRs have never investigated the number of pathways and compartments that we have included in our study. We have clarified the sentence to better describe this advance:

“This endeavor led to the most thorough investigation into the signaling capacity of any GPCR to date (15 pathways in 4 compartments).” (page 11)

- It would be helpful for the authors to provide confocal images demonstrating receptor localization pre and post ligand treatment rather than mGs. It is difficult to make comments about receptor trafficking with the current data.

As requested, we have now included confocal images of SNAP-tagged GLP-1R pre and post ligand treatment (**Supplementary Fig. 6**). The results clearly demonstrate that plasma membrane-localized GLP-1R traffics to early endosomes and further to the Golgi apparatus following stimulation with semaglutide. In parallel, we performed BRET experiments to qualitatively measure the basal localization of luciferase-tagged GLP-1R with acceptor-tagged compartment markers. These findings show that GLP-1R mainly localizes to the plasma membrane with smaller populations in the Golgi and ER followed by endosomes. In line with our other findings, stimulation of GLP-1R with semaglutide results in a decrease in BRET at the plasma membrane and an increase in BRET at early endosomes and the Golgi apparatus. We also observe a decrease in BRET at the ER which we interpret as a feedback mechanism for anterograde trafficking, but this requires further investigation (**Supplementary Fig. 5**).

- For all concentration-response curves in both main and supplemental figures, please include the time point following ligand stimulation for which these data are representing either in the figure or figure legend.

We have now included **Supplementary Table 10** that includes additional details regarding the transfection conditions as well as the ligand incubation times for every pathway and compartment.

- In Extended Data 5, 7, and 9, it would be helpful to have the coloring of the data when grouping by signaling pathway, be consistent with those depicted in the preceding figures (i.e. each ligand is given a specific color, and that same color is used when showing the data grouped by both ligands and by effector).

We have now recoloured the data according to the ligand colours assigned in other parts of the manuscript. The reviewer can find these new colouring scheme in **Supplementary Figures 3, 11, 14 and 17**.

.....

References

- 1 Pandy-Szekeres, G. *et al.* GPCRdb in 2018: adding GPCR structure models and ligands. *Nucleic Acids Res* **46**, D440-D446, doi:10.1093/nar/gkx1109 (2018).
- 2 Bekker-Jensen, D. B. *et al.* Rapid and site-specific deep phosphoproteome profiling by data-independent acquisition without the need for spectral libraries. *Nat Commun* **11**, 787, doi:10.1038/s41467-020-14609-1 (2020).
- 3 Humphrey, S. J., Karayel, O., James, D. E. & Mann, M. High-throughput and high-sensitivity phosphoproteomics with the EasyPhos platform. *Nat Protoc* **13**, 1897-1916, doi:10.1038/s41596-018-0014-9 (2018).
- 4 Leutert, M., Rodriguez-Mias, R. A., Fukuda, N. K. & Villen, J. R2-P2 rapid-robotic phosphoproteomics enables multidimensional cell signaling studies. *Mol Syst Biol* **15**, e9021, doi:10.15252/msb.20199021 (2019).
- 5 Martinez-Val, A. *et al.* Spatial-proteomics reveals phospho-signaling dynamics at subcellular resolution. *Nat Commun* **12**, 7113, doi:10.1038/s41467-021-27398-y (2021).

- 6 Salovska, B. *et al.* Phosphoproteomic analysis of metformin signaling in colorectal cancer cells elucidates mechanism of action and potential therapeutic opportunities. *Clin Transl Med* **13**, e1179, doi:10.1002/ctm2.1179 (2023).
- 7 Kitata, R. B. *et al.* A data-independent acquisition-based global phosphoproteomics system enables deep profiling. *Nat Commun* **12**, 2539, doi:10.1038/s41467-021-22759-z (2021).
- 8 Avet, C. *et al.* Effector membrane translocation biosensors reveal G protein and betaarrestin coupling profiles of 100 therapeutically relevant GPCRs. *Elife* **11**, doi:10.7554/eLife.74101 (2022).
- 9 Inoue, A. *et al.* Illuminating G-Protein-Coupling Selectivity of GPCRs. *Cell* **177**, 1933-1947 e1925, doi:10.1016/j.cell.2019.04.044 (2019).
- 10 Okashah, N. *et al.* Variable G protein determinants of GPCR coupling selectivity. *Proc Natl Acad Sci U S A* **116**, 12054-12059, doi:10.1073/pnas.1905993116 (2019).
- 11 Olsen, R. H. J. *et al.* TRUPATH, an open-source biosensor platform for interrogating the GPCR transducerome. *Nat Chem Biol* **16**, 841-849, doi:10.1038/s41589-020-0535-8 (2020).
- 12 Pandey, S. *et al.* Intrinsic bias at non-canonical, beta-arrestin-coupled seven transmembrane receptors. *Mol Cell* **81**, 4605-4621 e4611, doi:10.1016/j.molcel.2021.09.007 (2021).
- 13 Zhang, X. *et al.* Differential GLP-1R Binding and Activation by Peptide and Non-peptide Agonists. *Mol Cell* **80**, 485-500 e487, doi:10.1016/j.molcel.2020.09.020 (2020).
- 14 Demichev, V. *et al.* A time-resolved proteomic and prognostic map of COVID-19. *Cell Syst* **12**, 780-794 e787, doi:10.1016/j.cels.2021.05.005 (2021).
- 15 Nusinow, D. P. *et al.* Quantitative Proteomics of the Cancer Cell Line Encyclopedia. *Cell* **180**, 387-402 e316, doi:10.1016/j.cell.2019.12.023 (2020).
- 16 Saei, A. A. *et al.* ProTargetMiner as a proteome signature library of anticancer molecules for functional discovery. *Nat Commun* **10**, 5715, doi:10.1038/s41467-019-13582-8 (2019).
- 17 Huang, L., Zalkikar, J. & Tiwari, R. C. Likelihood ratio test-based method for signal detection in drug classes using FDA's AERS database. *J Biopharm Stat* **23**, 178-200, doi:10.1080/10543406.2013.736810 (2013).
- 18 Ochoa, D. *et al.* The next-generation Open Targets Platform: reimaged, redesigned, rebuilt. *Nucleic Acids Res* **51**, D1353-D1359, doi:10.1093/nar/gkac1046 (2023).

REVIEWERS' COMMENTS

[Reviewer #2 confidentially signed off on the manuscript.]

Reviewer #3 (Remarks to the Author):

Authors have addressed my concerns. While I don't completely agree with the approaches to quantifying bias, etc., I think the authors have used reasonable and thoughtful methods to assess it.